# "Mass gathering events and COVID-19 transmission in Borriana (Spain): A retrospective cohort study"

**Salvador Domènech-Montoliu[1], Maria Rosario Pac-Sa[2], Paula Vidal-Utrillas[3], Marta Latorre-Poveda[1], Alba Del Rio-González[3], Sara Ferrando-Rubert[3], Gema Ferrer-Abad[3], Manuel Sánchez-Urbano[1], Laura Aparisi-Esteve[4], Gema Badenes-Marques[1], Belén Cervera-Ferrer[1], Ursula Clerig-Arnau[1], Claudia Dols-Bernad[5], Maria Fontal-Carcel[6], Lorna Gomez-Lanas[1], David Jovani-Sales[4], Maria Carmen León-Domingo[7], Maria Dolores Llopico-Vilanova[1], Mercedes Moros-Blasco[4], Cristina Notari-Rodríguez[1], Raquel Ruíz-Puig[1], Sonia Valls-López[1], Alberto Arnedo-Pena[2,8,9]***

1 Emergency Service, Hospital de la Plana, Vila-real, Castellon, Spain, 2 Public Health Center, Castelló de la Plana, Castellon, Spain, 3 Health Centers I and II, Borriana, Castellon, Spain, 4 Carinyena Health Center, Vila-real, Castellon, Spain, 5 Health Center, Onda, Castellon, Spain, 6 Health Center, La Vall d'Uixó, Castellon, Spain, 7 Villa Fátima School, Borriana, Castellon, Spain, 8 Department of Health Science, Public University Navarra, Pamplona, Spain, 9 Epidemiology and Public Health (CIBERESP), Madrid, Spain

* albertoarnedopena@gmail.com

## Abstract

### Objective

Mass gathering events (MGEs) are associated with the transmission of COVID-19. Between 6 and 10 March 2020, several MGEs related to the *Falles* festival took place in Borriana, a municipality in the province of Castellon (Spain). The aim of this study was to estimate the incidence of COVID-19 and its association with these MGEs, and to quantify the potential risk factors of its occurrence.

### Methods

During May and June 2020, a population-based retrospective cohort study was carried out by the Public Health Center of Castelló and the Hospital de la Plana in Vila-real. Participants were obtained from a representative sample of 1663 people with potential exposure at six MGEs. A questionnaire survey was carried out to obtain information about attendance at MGEs and COVID-19 disease. In addition, a serologic survey of antibodies against SARS-Cov-2 was implemented. Inverse probability weighted regression was used in the statistical analysis.

### Results

A total of 1338 subjects participated in the questionnaire survey (80.5%), 997 of whom undertook the serologic survey. Five hundred and seventy cases were observed with an attack rate (AR) of 42.6%; average age was 36 years, 62.3% were female, 536 cases were confirmed by laboratory tests, and 514 cases were found with SARS-CoV-2 total antibodies.

**Data Availability Statement:** All relevant data are within the manuscript and its Supporting Information files.

**Funding:** The author(s) received no specific funding for this work.

**Competing interests:** The authors have declared that no competing interest exist.

Considering MGE exposure, AR was 39.2% (496/1264). A dose-response relationship was found between MGE attendance and the disease, (adjusted relative risk [aRR] = 4.11 95% confidence interval [CI]3.25–5.19). Two MGEs with a dinner and dance in the same building had higher risks. Associated risk factors with the incidence were older age, obesity, and upper and middle class versus lower class; current smoking was protective.

## Conclusions

The study suggests the significance of MGEs in the COVID-19 transmission that could explain the subsequent outbreak in Borriana.

## Introduction

Mass gathering events (MGEs) are important risk factors of severe acute respiratory coronavirus 2 (SARS-CoV-2) transmissions, which causes coronavirus disease 2019 (COVID-19) pandemic [1]. According to the World Health Organization [2] "Mass gatherings are events characterized by the concentration of people at a specific location for a specific purpose over a set period of time that have the potential to strain the planning and response resources of the host country or community". MGEs cover different types of event and contexts such as public and private celebrations, festivals, religious events and pilgrimages, sporting and touristic events, and political meetings. The crucial role of MGEs in the global propagation of the disease has been evidenced in several countries, including China [3], Iran [4], Malaysia [5], Italy [6], Spain [7], France [8], Germany [9], Jordan [10], Malta [11], Switzerland [12], and Malawi [13]. Significant international efforts have been made to implement specific measures, risk assessment and surveillance, and event cancellations in order to prevent the spread of SARS-CoV-2 from MGEs [14–17]. The propagation of SARS-CoV-3 in these MGEs was measured with the basic reproductive number (Ro), which reflects the efficiency of transmission of the disease [18], and "is defined as the expected number of secondary cases produced by a single (typical) infection in a completely susceptible population" [19]. A meta-analysis of medical literature estimated a Ro = 3.38±1.40 from a range of 1.90–6.49 for the COVID-19 pandemic [20].

MGEs imply the gathering of people in restricted spaces, either indoor or outdoor, over a prolonged period of time, where food and/or drink are generally consumed, usually in close proximity to others, and involving the movement of populations [14, 21–23]. The conditions of MGEs have been associated with the spread of SARS-CoV-2, but few MGE studies [24, 25] have published quantification and adjustment for potential risk factors. In addition, several epidemiologic biases such as selection and misclassification have been observed in some studies of SARS-CoV-2 outbreaks, considering the novelty of the disease and the emergency situation [26].

Spain has had a high incidence of COVID-19 [27, 28], with a large number of COVID-19 outbreaks occurring in households, nursing homes, hospitals, workplaces and leisure facilities [29–33]. We studied a MGE COVID-19 outbreak that took place in the first wave of the pandemic. During March and April 2020 in Borriana, a municipality with 34,683 inhabitants located in the province of Castellon in the Valencian Community (Spain), a COVID-19 outbreak occurred with an incidence of 260 cases (749.6 cases per 100,000 inhabitants) confirmed by reverse-transcriptase polymerase chain reaction (RT-PCR) [34]. Between February and the first days of March 2020, before the COVID-19 outbreak in Borriana, several MGEs took place

in connection with the traditional *Falles* festival, which is held annually in Borriana. Our hypothesis was that the MGEs held during the *Falles* festival were associated with the COVID-19 outbreak in Borriana.

Using a population-based retrospective cohort study, we aimed to estimate the association of the incidence of COVID-19 disease with the MGEs in Borriana, and quantify potential risk factors of its occurrence.

## Material and methods

### Description of MGEs during the *Falles* festival in Borriana

Borriana is a municipality located 5.7 km from the Mediterranean Sea in the province of Castellon, Spain. A series of MGEs took place between March 6 and 10, 2020 during the traditional *Falles* festival in Borriana. This popular festival is organized by people in the town's different neighborhoods, clustered in social groups, known as a "*falla*" (singular) or "*falles*" (plural), with the purpose of bringing the festivities to the streets. *Falles* groups consider themselves as a large family. Their final objective is to build a monument with humorous scenes of daily life, which is burned on March 19 [35, 36]. During the 2020 festivities, 19 *falles* with a total of 2800 members were active in Borriana; each group had between 26 and 384 people and a median of 143 members. The members of the *falles* comprise around 8.1% of Borriana's population.

The MGEs analyzed in this study took place in three locations: building A, purposely designed for MGEs with a surface area of 1670 m$^2$ and a capacity of 900 people (three events); theater B (two events and a capacity of 884), and an outdoor square in the city of Valencia. Building A and theater B had air conditioning and ventilation equipment. The MGEs are described below:

First, a *pa-i-porta* ('bring your own' supper) (building A; March 6), a community dinner with an estimated total attendance of approximately 1400 people over a seven hour period (9:30 p.m. to 4:30 a.m.) and dancing after midnight.

Second, the Queen's gala dinner (building A; March 7), with 400 people gathered together over six and a half hours (10:00 p.m. to 4:30 a.m.) and dancing after midnight. A detailed list of attendees and their distribution in building A was available.

Third, a trip to see fireworks in a square in Valencia (March 8). About 800 people gathered for half an hour (at 2:00 p.m.).

Fourth, a senior citizens' dance (building A; March 8). Around 100 people attended for one and a half hours (5:30 p.m. to 7:00 p.m.).

Fifth, the theater awards gala (theater B; March 8): indoor show attended by 300 people for two hours (7:00 p.m. to 9:00 p.m.).

Sixth, the Queen's offering (theater B; March 10): a show attended by 400 people for one and a half hours (10:30 p.m. to 0:00 a.m.).

A summary of the characteristics of these MGEs is presented in Table 1.

### Design of the study

A population-based retrospective cohort study was carried out from May 14 to June 31, 2020 in Borriana. The study was jointly designed by the Public Health Center of Castellon and the Emergency Service of the Hospital de la Plana (HP) in Vila-real. The study had two phases: 1) a survey with a specific questionnaire to estimate the incidence of COVID-19, and 2) a serologic study of antibodies against SARS-CoV-2 to confirm cases with laboratory tests and to uncover the extent of the infection and the epidemic situation.

**Table 1. Characteristics of Mass Gathering Events (MGEs) connected to the *Falles* festival in Borriana from March 6 to 10.**

| MGEs | Location | Date | Hours | People in attendance | Activities |
|---|---|---|---|---|---|
| *Pa-i-porta* supper | Building A | 3/6/2020 | 7.00 | 1400[1] | Dinner and dance |
| Queen's gala dinner | Building A | 3/7/2020 | 6.30 | 400 | Dinner and dance |
| Trip to Valencia | Square outdoor | 3/8/2020 | 0.30 | 800 | Attendance |
| Senior citizen's dance | Building A | 3/8/2002 | 1.30 | 100 | Dance |
| Theater awards gala | Theater B | 3/8/2020 | 2.00 | 300 | Attendance |
| Queen's offering | Theater B | 3/10/2020 | 1.30 | 400 | Attendance |

1. Throughout the duration of the MGE.

## Questionnaire survey

In the first phase, starting on May 14, 2020, a standardized questionnaire was administered to obtain information about MGE exposure, demographic characteristics, occupation, habits, health condition, symptoms of the disease, the evolution of COVID-19 disease, previously conducted COVID-19 laboratory tests, and family members affected by COVID-19. The study period covering COVID-19 cases ranged from January 1 to June 31, 2020. The questionnaire was completed through a telephone survey carried out by the health staff of HP, Borriana health centers, and other health centers in the Health Department of La Plana in Vila-real, Castellon. In the telephone survey, parents were requested to ask the questions to their children when a child had been chosen in the simple random sampling. In general, parents were able to answer the questionnaire. Social class was estimated from occupations; children's social class was that of their parents. Two groups were considered: I and II included professional, managerial and technical occupations (upper and middle class); group III-VI included skilled, non-manual or manual, semi-skilled, and unskilled occupations (lower class).

Participants were recruited from two sources. First, a representative sample of the 19 *falles* (n = 2800 members) was obtained by simple random sampling with design effect 1 and 19 clusters (one per *falla)*, considering an attack rate of COVID-19 disease of 50% in MGEs, a confidence level of 80%, and an alpha error 5%. The sample comprised 1558 people, which corresponds to 82 people per *falla* for those with a population larger than 82, or all the members of the *falla* for those with a population smaller than 82. The Open-Epi program [37] was used to randomly select the participants from a numbered list of all members of each *falla* until 82 people, including adults and children, were obtained. The phone numbers of each selected participant or their family member were then obtained from the Borriana *Falles* organization. Second, in order to maximize the number of participants from the Queen's gala dinner (n = 400), given that the complete list of attendees to the event was available, an additional random sample of 105 people from the list of diners was recruited. The same procedure described above was used to randomly select the 105 people.

## Serologic survey and laboratory tests

During the second phase, a serologic survey of anti-SARS-CoV-2 antibody prevalence was carried out from 23 to 27 June 2020 by the Clinical Analysis and Microbiology Service (CAMS) of HP. Qualitative detection of antibodies against SARS-CoV-2 was carried out by an electrochemiluminescence immunoassay (ECLIA) (Elecsys® Anti-SARS-CoV-2, Roche Diagnostics) [38]. IgG and IgM antibodies against SARS-CoV-2 were detected by a lateral flow immunochromatographic assay (LFIC) (Healgen Scientific LLC for COVID-19 IgG/IgM rapid test cassette) [39].

Additionally, information about previously conducted laboratory tests for COVID-19 was gathered during the questionnaire survey. These data consisted of 1) RT-PCR tests, including LightMix® Modular Sarbecovirus E-gene with the LightCycler® 480 II system (Roche, Basel, Switzerland) [40], 2) ECLIA with LFIC, and 3) rapid anti-SARS-CoV-2 antibody tests. These tests were conducted by HP and other public and private laboratories.

## Case definitions

A probable COVID-19 case was defined as a patient who presented clinical and epidemiological criteria of COVID-19 disease during the study period, according to the case-definition proposed by the Center for Disease Control and Prevention [41]. Clinical and epidemiological criteria must fulfill two conditions. The first is the reporting of at least two of the following symptoms: fever, chills, rigors, myalgia, headache, sore throat, and new olfactory and taste disorder; or one of the following symptoms: cough, shortness of breath, or difficulty breathing; or severe respiratory illness such as pneumonia or acute respiratory distress syndrome. The second condition is contact with a confirmed COVID-19 patient, or living in an area where a COVID-19 outbreak has been reported.

A confirmed COVID-19 case was defined as a patient who had positive antibodies of SARS-CoV-2 by ECLIA with LFIC, positive PCR, or positive rapid anti-SARS-CoV-2 antibody tests during the study period.

Non-cases (negative cases) were defined as participants who had no COVID-19 clinical criteria or had negative COVID-19 test results during the study period.

## Statistical methods

The program Epi-Info® version 7 [42] was used to calculate the sample size. COVID-19 attack rates (AR) were estimated as the quotient between cases and total exposed population, considering different variables. To estimate the risk of the MGEs, COVID-19 cases with onset of symptoms between March 6 and 31, 2020 were included. The *pa-i-porta* and Queen's gala dinner MGE events were analyzed in greater detail than the other events, considering the attendance population and the potential risk of COVID-19. Chi2 and Fisher's exact test were used in the comparisons among variables.

Associations of risk factors with COVID-19 were measured by the relative risk (RR) using Poisson regression and multilevel Poisson regression, considering *falles* as a level group with a 95% confidence interval (CI). In order to adjust for potential confounding factors, the directed acyclic graphs (DAGs) method [43] was used with the DAGitty® program (version 3.0) [44], together with inverse probability weighted regression [45] to obtain adjusted AR (aAR) and RR (aRR). The DAG provides a picture of the relationship between an exposure (mass gathering event) and an outcome (COVID-19 disease) and factors that could be potential confounders. An adjustment was made for these factors. The factors were age, sex, social class (upper and middle class versus lower class), chronic illness, family COVID-19 case and *falla* (social group); all these factors could alter the relationship between exposure and outcome. In addition, a sensitivity analysis was carried out including participants who were tested for COVID-19 disease, to gain more specificity of the results. The Stata ® program (version 14) [46] was used in the calculation.

## Ethical aspects

The study was approved by the director of the Public Health Center of Castellon and the management of the Health Department of La Plana, taking into account the situation of the COVID-19 pandemic in the province of Castellon. Participation was voluntary, and after an

explanation of the study objective, oral informed consent was obtained from all participants (or their parents, in the case of minors) included in the study. To ensure explicit consent, several clarifications were made before administering the survey questionnaire, including voluntary participation, the option to withdraw from the study without prejudice, the opportunity to receive information about the study, a guarantee that personal information would be kept in the strictest confidentiality and that privacy would be safeguarded as the data collected would be used anonymously for scientific research; the researcher responsible for data collection was identified. When a child was selected in the sampling process, permission to participate was requested from their parents, who would also help them to answer the questionnaire; all the above clarifications were also made before the survey questionnaire was administered.

## Results

### Description of participants in the questionnaire survey

Participation rates in the questionnaire survey were 80.5% (1338/1663) (Fig 1). Participation was higher among females than males (58.9% versus 41.1%). The mean age was 33.9 ± 17.8 years (rank of 1–80 years) (Table 2). The most represented age groups were 45–64 years (28.2%) and 15–24 years (20.1%), and the least represented groups were 1–4 years (2.3%) and 65 years and above (3.4%) (Table 2). The occupation III-VI group (lower class) was higher than the I-II group (upper and middle classes) (74.3% versus 25.7%) (Table 2). Chronic illness was present in 30.9% of participants. The *pa-i-porta* event was the most highly attended MGE (60.5%), followed by the Queen's gala dinner (24.8%). The theater award gala (12.4%) and the senior citizens' dance (2.8%) had the lowest attendances (Table 2). A total of 397 participants in the questionnaire survey (25.3%) did not attend any MGEs.

### Description of participants in the serologic survey and COVID-19 laboratory tests

A total of 1132 participants (84.6%) underwent laboratory tests for COVID-19 disease (Table 2), considering both the tests carried out as part of the serologic survey (n = 997) and those from private and public medical laboratories (n = 135) (Fig 1, see Methods for further information). Participation rates in the serologic survey were 74.5% (997/1338). Female participation in laboratory tests for COVID-19 (including both the serologic survey and medical records) was higher (60.1%) than that of males (39.9%). The mean age of participants that underwent a COVID-19 laboratory test was 36.9 ± 17.1 years. The 45–64 years and 15–24 years age groups were the most highly represented (30.8% and 19.2%, respectively), while 0–4 years (1.8%) and 65 years and above (3.4%) were the least represented (Table 2). Participation of the occupation III-VI group (lower class) was higher than in the I-II group (upper and middle class) (74.2% versus 25.0%) (Table 2). A total of 32.2% of the participants reported suffering from a chronic illness (Table 2). MGE attendance among participants with COVID-19 laboratory tests showed the *pa-i-porta* event as the most highly attended event (65.4%), again followed by the Queen's gala dinner (28.9%), while the theater award gala (13.3%) and the senior citizens' dance (3.4%) had the lowest attendance (Table 2). A total of 274 participants (24.1%) with COVID-19 laboratory tests did not attend any MGEs (Table 2).

### Description of COVID-19 outbreak

During the study period, 570 COVID-19 cases were found (Fig 1, Table 2). Among these, 536 (94.0%) were confirmed cases with positive COVID-19 laboratory tests and 34 (6.0%) were probable cases as diagnosed according to clinical and epidemiological criteria (Fig 1).

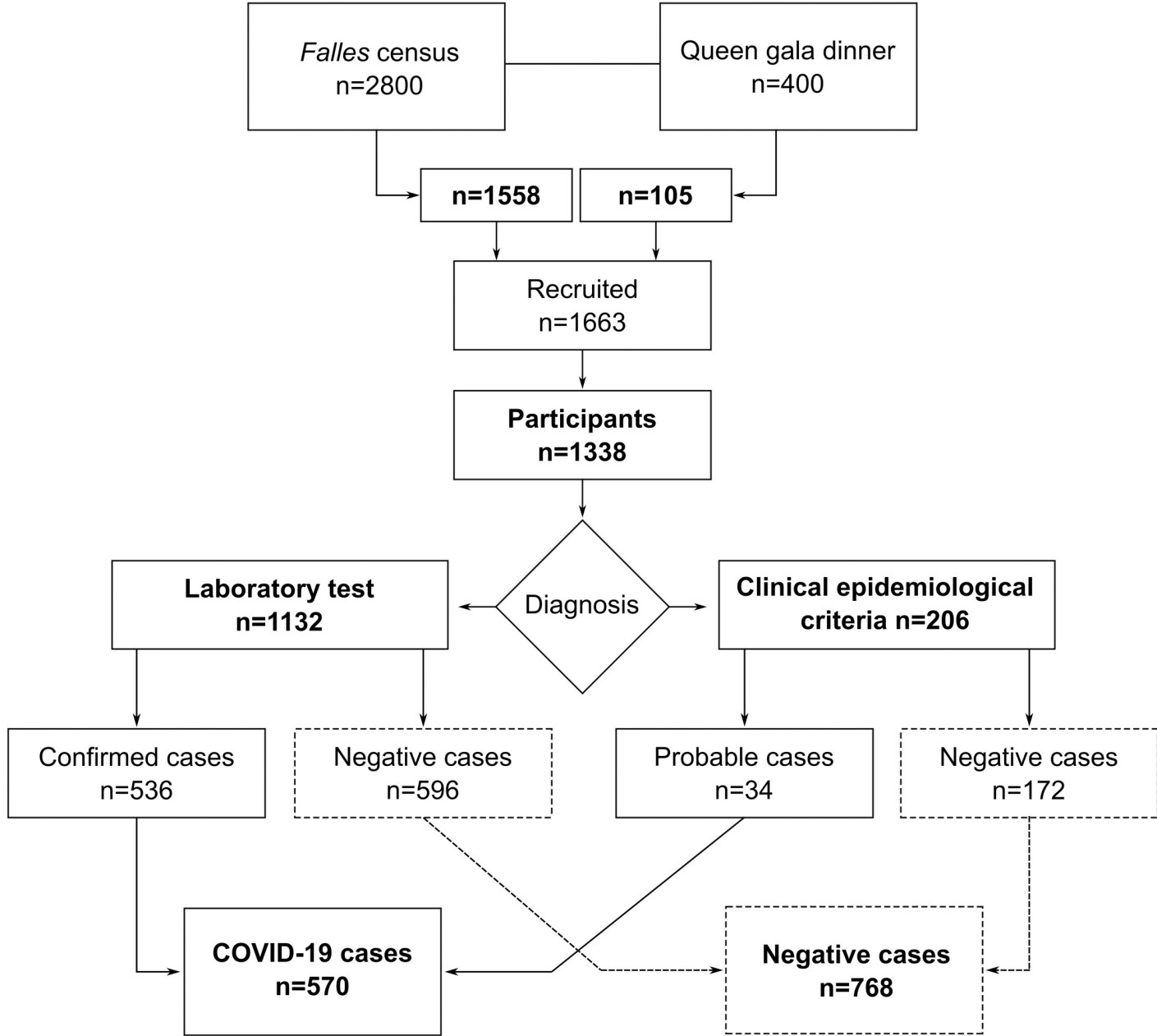

**Fig 1. Flow diagram of the study population in the MGEs in Borriana between January and June 2020.** COVID-19 cases included those diagnosed by laboratory tests (confirmed cases) or by clinical and epidemiological criteria (probable cases). Cases confirmed by laboratory tests included participants with positive results from the serologic survey as well as from private and public medical laboratories.

Specifically, confirmed cases included 514 participants (90.2%) with positive total anti-SARS-CoV-2 antibodies (32 of whom also reported a positive PCR test), seven participants (1.2%) with positive PCR only (a total of 39 participants had a positive PCR, 6.8%), and 15 participants (2.6%) with positive rapid anti-SARS-CoV-2 antibody tests. From the subset of 997 participants in the serologic survey, 508 were positive for anti-SARS-CoV-2 IgG antibodies, and eight participants were also positive for anti-SARS-CoV-2 IgM antibodies. AR was 42.6% (570/1338) among total participants, 47.3 (536/1132) among participants undergoing

**Table 2. Characterization of the participants included in the study of Mass Gathering Events (MGEs) from January 1 to June 31, 2020, Borriana.**

| Variable | Participants (n) | Participants1 (%) | COVID-19 tests (n) | COVID-19 tests (%) |
|---|---|---|---|---|
| Total participants | 1338 | - | 1132 | 84.61 |
| Female | 788 | 58.9 | 680 | 60.1 |
| Male | 550 | 41.1 | 452 | 39.9 |
| Mean age ± SD2 | 33.9 ± 17.8 | - | 36.9 ± 17.1 | - |
| Age (years) | | | | |
| 0–4 | 31 | 2.3 | 20 | 1.8 |
| 5–14 | 190 | 14.2 | 137 | 12.1 |
| 15–24 | 269 | 20.1 | 217 | 19.2 |
| 25–34 | 185 | 13.8 | 157 | 13.9 |
| 35–44 | 241 | 18.0 | 213 | 18.8 |
| 45–64 | 377 | 28.2 | 349 | 30.8 |
| 65 and above | 45 | 3.4 | 39 | 3.4 |
| Social class[3,4] | | | | |
| Occupation I-II | 339 | 25.7 | 283 | 25.0 |
| Occupation III-VI | 978 | 74.3 | 840 | 74.2 |
| Chronic illness[5] | | | | |
| Yes | 411 | 30.9 | 364 | 32.2 |
| No | 920 | 69.1 | 763 | 67.4 |
| MGE total attendees | | | | |
| *Pa-i-porta* (n = 1400) | 809 (57.8%) | 60.5 | 740 | 65.4 |
| Queen's gala dinner (n = 400) | 332 (83.0%) | 24.8 | 317 | 28.0 |
| Valencia trip (n = 800) | 211 (26.4%) | 15.8 | 190 | 16.8 |
| Queen's offering (n = 400) | 239 (59.8%) | 17.9 | 230 | 20.3 |
| Senior citizens' dance (n = 100) | 38 (38.0%) | 2.8 | 38 | 3.4 |
| Theater awards gala (n = 300) | 166 (55.3%) | 12.4 | 151 | 13.3 |
| No attendance | 397 | 25.3 | 274 | 24.2 |

1. Of total participants.

2. Standard deviation.

3. Occupation group I and II: Professional, managerial and technical occupations (upper and middle class); Group III-VI: Skilled, non-manual or manual; semi-skilled; unskilled occupations (lower class).

4. Missing answer from 21 participants.

5. Missing answer from 7 participants.

laboratory tests, and 16.5% (34/206) among those classified according to clinical and epidemiological criteria (Fig 1).

The temporal distribution of confirmed and probable COVID-19 cases is shown by onset of symptoms (Fig 2). The first cases were reported on 28 and 29 January 2020. The incidence then slowly progressed during February with a small peak of cases on March 2. Two maximum peaks of cases occurred on March 9 (45 cases) and March 16 (49 cases). After April, only isolated cases were reported, which is consistent with the general lockdown in Spain that was implemented on March 15. According to these observations, the distribution of COVID-19 cases in this study showed a bimodal epidemic curve with two maxima. Interestingly, the aforementioned peaks took place between 3 and 10 days after the MGEs (see Methods for further information). This indicates several point sources of the outbreak, with the *pa-i-porta* and Queen's gala dinner events being the first ones on March 6 and 7, respectively.

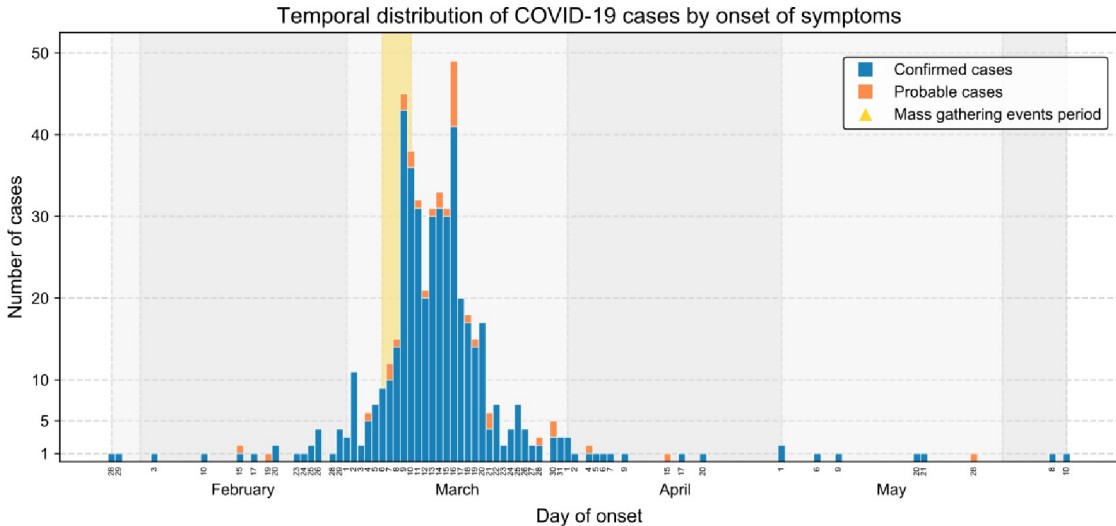

**Fig 2. Temporal distribution of COVID-19 cases by onset of symptoms.** The numbers of confirmed and probable COVID-19 cases are shown in blue and orange, respectively. The yellow vertical line highlights the time period when the six MGEs took place (March 6–10, 2020).

**Epidemiological characterization of COVID-19 cases.** Regarding the sex of total cases, 355 (62.3%) were female and 215 (37.7%) were male (Table 3). The AR for males (39.1%) was smaller than for females (45.1%) (RR = 0.87, 95% CI 0.73–1.03) (Table 2). Similarly, COVID-19 incidence was lower in males than in females for both confirmed (44.9% versus 49.0%) and probable cases (12.2% versus 20.4%), although differences in AR were not significant.

Regarding age, cases were older than non-cases (36.4 ± 17.1 years versus 32.0 ± 18.0 years; p<0.001) (Table 3). When split by age groups, the highest ARs were found in the 55–64 years group (50.1%) and the 35–44 years group (47.3%), whereas the lowest ARs were observed in the 0–4 years group (25.8%) and 5–14 years group (30.5%) (Table 3), but no significant differences were observed among age groups when models were adjusted for *falla*.

Regarding the clinical presentation of the disease, a total of 503 cases (88.2%), including 469 confirmed cases and 34 probable cases, showed COVID-19 illness. The signs and symptoms reported by COVID-19 patients included weakness (56.3%), fever (55.1%), loss of smell and/or taste (53.8%), myalgia (51.3%), headache (46.2%) cough (49.0%), sore throat (35.7%), coryza (31.8%), diarrhea (26.6%), dermatologic lesions (12.1%), vomiting (5.7%) dyspnea (4.6%) and pneumonia (2.4%). The average disease duration was 16.1 ± 20.9 days, with a median of 7.0 days (rank 1–100). Long-term symptoms or aftermaths of COVID-19 were reported in 6.6% of cases. Of note, 247 cases (43.4%) sought medical assistance and 13 cases (2.3%) required hospitalization due to COVID-19. Only one death attributable to COVID-19 was reported during the period of the study. On the other hand, asymptomatic cases made up 12.5% of all confirmed cases (67/536), with an average age of 25.2 ± 17.6 years. No differences in the number of asymptomatic cases were found between males and females. Among total cases, 33.5% reported having a chronic disease, the most frequent of which were cardiovascular disease, hypertension, allergic rhinitis, and hypothyroidism (Table 3). No significant associations with COVID-19 incidence were found in any of the analyzed diseases. Finally, the effect of having a family member with COVID-19 at the time of disease onset was analyzed. AR among those participants who reported having a COVID-19 positive family member was 57.7% (347/601). In contrast, AR among participants who did not report a COVID-19 positive family member

**Table 3. Characteristics of patients of COVID-19.** From January 1 to June 31, 2020, Borriana.

| Variables | Cases | Non-cases | Total | AR[1] (%) | RR (95% IC)[2] | p-value |
|---|---|---|---|---|---|---|
| Total cases | 570 | 768 | 1338 | 42.6 | - | - |
| PCR positive | 39 | 1299 | 1338 | 2.9 | - | - |
| Hospitalizations | 13 | 1325 | 1338 | 1.0 | - | - |
| Deaths | 1 | 1337 | 1338 | 0.07 | - | '- |
| Sex | | | | | | |
| Male | 215 | 335 | 550 | 39.1 | 0.87(0.73–1.03) | 0.110 |
| Female | 355 | 433 | 788 | 45.1 | 1.00 | |
| Age mean ± SD[3] | 36.4±17.1 32.0±18.0 | | | | 1.01 (1.00–1.01) | 0.001 |
| Age groups (years) | | | | | | |
| 0–4 | 8 | 23 | 31 | 25.8 | 1.00 | |
| 5–14 | 58 | 133 | 190 | 30.5 | 1.11(0.56–2.44) | 0.685 |
| 15–24 | 110 | 159 | 269 | 41.1 | 1.54(0.75–3.17) | 0.243 |
| 25–34 | 72 | 113 | 185 | 38.9 | 1.48(0.71–3.08) | 0.293 |
| 35–44 | 114 | 127 | 241 | 47.3 | 1.80(0.88–3.68) | 0.110 |
| 45–64 | 189 | 188 | 377 | 50.1 | 1.92(0.94–3.90) | 0.073 |
| 65 and above | 19 | 26 | 45 | 42.2 | 1.56(0.68–3.57) | 0.297 |
| Confirmed cases | | | | | | |
| Male | 203 | 249 | 452 | 44.9 | 0.91(0.77–1.09) | 0.322 |
| Female | 333 | 347 | 680 | 49.0 | 1.00 | |
| Probable cases | | | | | | |
| Male | 12 | 86 | 98 | 12.2 | 0.63(0.30–1.28) | 0.200 |
| Female | 22 | 86 | 108 | 20.4 | 1.00 | |
| Asymptomatic | | | | | | |
| Male | 30 | 520 | 550 | 5.5 | 1.17(0.72–1.90) | 0.519 |
| Female | 37 | 751 | 788 | 4.7 | 1.00 | |
| Family with COVID-19[4]    Yes | 347 | 254 | 601 | 57.7 | 1.91(1.59–2.23) | <0.001 |
| No | 221 | 506 | 727 | 30.4 | 1.00 | |
| Medical assistance    Yes | 247 | 94 | 341 | 72.4 | 2.22(1.88–2.62) | <0.001 |
| No | 323 | 674 | 997 | 32.1 | 1.00 | |
| Chronic illness[5]    Yes | 189 | 222 | 411 | 46.0 | 1.13(0.95–1.35) | 0.183 |
| No | 375 | 545 | 920 | 40.8 | 1.00 | |
| Diabetes Mellitus[6]    Yes | 10 | 18 | 28 | 35.7 | 0.84(0.45–1.56) | 0.574 |
| No | 552 | 743 | 1295 | 42.6 | 1.00 | |
| Cardiovascular diseases[6]Yes | 69 | 66 | 135 | 51.1 | 1.23(0.95–1.58) | 0.114 |
| No | 493 | 695 | 1188 | 41.5 | 1.00 | |
| Hypertension[6]    Yes | 47 | 51 | 98 | 48.0 | 1.14(0.84–1.54) | 0.393 |
| No | 514 | 711 | 1225 | 42.0 | 1.00 | |
| Hypothyroidism[7]    Yes | 28 | 22 | 50 | 56.0 | 1.32(0.90–1.94) | 0.150 |
| No | 534 | 740 | 1274 | 41.9 | 1.00 | |
| Digestive diseases[6]    Yes | 16 | 15 | 31 | 51.6 | 1.21(0.74–2.00) | 0.450 |
| No | 545 | 747 | 1292 | 42.2 | 1.00 | |
| Asthma[7]    Yes | 11 | 27 | 38 | 29.0 | 0.66(0.36–1.20) | 0.174 |
| No | 551 | 735 | 1286 | 42.8 | 1.00 | |
| Allergic Rhinitis[7]    Yes | 27 | 31 | 58 | 46.6 | 1.11(0.75–1.62) | 0.613 |
| No | 535 | 731 | 1266 | 42.3 | 1.00 | |

1. AR = attack rate.

2. Adjusted for falla.

3. SD = Standard deviation.

4. Missing answers from 10 participants.

5. Missing answers from 7 participants.

6. Missing answers from 15 participants.

7. Missing answers from 14 participants.

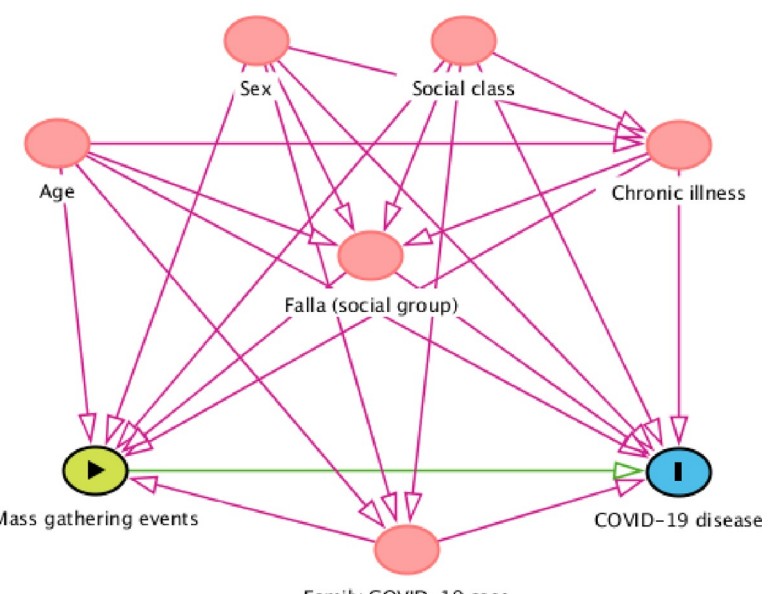

**Fig 3. Directed Acyclic Graphs (DAG) of mass gathering events (exposure) effect on COVID-19 disease (outcome).** Ancestors of exposure and outcome (in red). Based on DAGitty version 3.0.

was 30.4% (221/727). Therefore, having a family member with COVID-19 increased the risk of COVID-19 incidence (RR = 1.91, 95% CI 1.59–2.23) (Table 3).

## Analysis of COVID-19 outbreak and MGEs

**General analysis of MGEs.** Considering the dates of the MGEs (6 and 10 March 2020) and the temporal distribution of COVID-19 cases in the study (Fig 2), the analysis focused further on COVID-19 cases attributable to the MGEs associated with the *Falles* in Borriana. The 74 cases that presented the onset of symptoms or a positive test obtained before March 6 or after March 31 were therefore excluded since their illness onset (before the MGEs or more than three weeks after the last MGE) could not be related to the studied MGEs. Consequently, 1264 participants were included in the analysis with 496 cases, and AR of 39.2% (496/1264). DAGs were used to study MGEs (exposure) and COVID-19 disease (outcome) and potential confounding factors (Fig 3). Raw AR and adjusted (aAR), as well as RR and aRR, of MGEs are presented in Table 4. The aRR of males and females did not differ significantly. The aRR increased with age, from 0–4 years (aAR = 17.0%) to 35–44 years, which presented the highest values (aAR = 46.9%) (aRR = 2.77 95% CI 1.30–5.88). The age groups of 45–64 years and 65 years and above also presented high values (aAR = 44.2%) (aRR = 2.60 95% CI 1.22–5.52) and (aAR = 39.2%) (aRR = 2.31 95% CI 1.07–4.99), respectively. In addition, COVID-19 disease was associated with occupations I-II (upper and middle class) (aRR = 1.22 95% CI 1.06–1.40).

Those who had a family member with COVID-19 had a higher risk of SARS-CoV-2 infection (aRR = 1.71 95% CI 1.49–1.97). Additionally, the risk of contracting COVID-19 increased with higher body mass index (BMI), up to the group of 30 Kg/m$^2$ or higher (aRR = 3.21 95% CI 1.29–7.98), and marginally with habitual alcoholic beverage consumption (aRR = 1.14 95% CI 0.99–1.32). Chronic illness and physical exercise were not associated with contracting COVID-19. On the other hand, the current smoker group presented lower risk (aRR = 0.63 95% CI 0.52–0.78). Regarding MGEs, the aAR ranged from 62.0% (the Queen's offering) to 47.9% (the Valencia trip). The disease was associated with the *pa-i-porta* event with an aRR of

**Table 4. Mass Gathering Event (MGE) attendance, Attack Rate (AR), adjusted AR (aAR), Relative Risk (RR), and adjusted (aRR).** Confidence interval (CI). From March 6 to 31, 2020, Borriana.

| Variables | | Cases | Non-cases | AR (%) | RR (95% IC) | aAR[1] (%) | aRR (95% IC)[1] | p-value |
|---|---|---|---|---|---|---|---|---|
| Male | | 189 | 335 | 36.1 | 0.87(0.73–1.04) | 39.2 | 0.98(0.86–1.16) | 0.785 |
| Female | | 307 | 433 | 41.5 | 1.00 | 39.9 | 1.00 | |
| Age 0–4 (years)[2] | | 6 | 23 | 20.7 | 1.00 | 17.0 | | |
| 5–14 | | 49 | 132 | 27.1 | 1.31(0.56–3.05) | 25.6 | 1.51(0.69–3.31) | 0.305 |
| 15–24 | | 93 | 159 | 36.7 | 1.77(0.78–4.05) | 35.9 | 2.12(0.99–4.54) | 0.054 |
| 25–34 | | 64 | 113 | 36.2 | 1.74(0.76–4.04) | 38.1 | 2.25(1.04–4.83) | 0.039 |
| 35–44 | | 106 | 127 | 45.5 | 2.20(0.97–5.00) | 46.9 | 2.77(1.30–5.88) | 0.008 |
| 45–64 | | 161 | 188 | 46.1 | 2.23(0.99–5.04) | 44.2 | 2.60(1.22–5.52) | 0.013 |
| 65 and above | | 18 | 26 | 40.9 | 1.97(0.78–4.98) | 39.2 | 2.31(1.07–4.99) | 0.032 |
| Occupation I-II[1] | | 153 | 166 | 48.0 | 1.30(1.08–1.58) | 45.7 | 1.22(1.06–1.40) | 0.005 |
| Occupation III-VI | | 340 | 585 | 36.8 | 1.00 | 37.6 | 1.00 | |
| Chronic illness[1] | Yes | 162 | 222 | 42.3 | 1.12(0.93–1.36) | 39.7 | 1.01(0.88–1.17) | 0.882 |
| | No | 328 | 545 | 37.6 | 1.00 | 39.3 | 1.00 | |
| Family with COVID-19 | Yes | 305 | 254 | 54.6 | 2.01 (1.67–2.41) | 51.3 | 1.71(1.49–1.97) | <0.001 |
| | No | 189 | 506 | 27.2 | | 29.9 | 1.00 | |
| Body Mass Index[1,3] <18.5 Kg/m[2] | | 12 | 31 | 27.9 | 1.00 | 17.5 | 1.00 | |
| 18.5–24.9 Kg/m[2] | | 203 | 327 | 38.3 | 1.32(0.96–1.80) | 41.5 | 2.37(0.96–5.85) | 0.062 |
| 25.0–29.9 Kg/m[2] | | 143 | 177 | 44.7 | 1.57(1.13–2.17) | 45.2 | 2.58(1.04–6.40) | 0.040 |
| ≥30.0 Kg/m[2] | | 81 | 74 | 52.3 | 1.68(1.18–2.40) | 56.3 | 3.21(1.29–7.98) | 0.012 |
| Physical exercise[1 3] | Yes | 258 | 401 | 39.2 | 0.89(0.74–1.06) | 40.1 | 0.88(0.77–1.01) | 0.077 |
| | No | 183 | 211 | 46.5 | 1.00 | 45.3 | 1.00 | |
| Drink alcohol[1 3] | Yes | 109 | 138 | 44.1 | 1.13(0.91–1.40) | 46.9 | 1.14(0.99–1.32) | 0.065 |
| | No | 332 | 473 | 41.2 | 1.00 | 40.9 | 1.00 | |
| Current smoker[1 3] | | 75 | 197 | 27.6 | 0.66(0.51–0.85) | 28.7 | 0.63(0.52–0.78) | <0.001 |
| Ex smoker[1 3] | | 108 | 100 | 51.9 | 1.25 (1.00–1.59) | 50.8 | 1.13 (0.97–1.31) | 0.122 |
| Never smoker[1 3] | | 256 | 311 | 45.9 | 1.00 | 45.0 | 1.00 | |
| Pa-i-porta[2] | Yes | 386 | 367 | 51.3 | 2.38(1.93–2.94) | 49.9 | 2.15(1.80–2.56) | <0.001 |
| | No | 110 | 401 | 21.5 | 1.00 | 23.4 | 1.00 | |
| Queen's gala dinner[2] | Yes | 198 | 110 | 64.3 | 2.06(1.72–2.47) | 61.9 | 1.90(1.67–2.17) | <0.001 |
| | No | 298 | 658 | 31.2 | 1.00 | 32.5 | 1.00 | |
| Valencia trip[2] | Yes | 92 | 105 | 46.7 | 1.23(0.98–1.55) | 47.9 | 1.25(1.06–1.46) | 0.007 |
| | No | 404 | 663 | 37.9 | 1.00 | 38.5 | 1.00 | |
| Queen's offering[2] | Yes | 139 | 74 | 65.3 | 1.92(1.58–2.34) | 62.0 | 1.77(1.54–2.04) | <0.001 |
| | No | 357 | 694 | 34.0 | 1.00 | 35.0 | 1.00 | |
| Senior citizens' dance[2] | Yes | 23 | 8 | 74.2 | 1.93(1.27–2.94) | 57.3 | 1.48(1.14–1.93) | 0.004 |
| | No | 473 | 760 | 38.4 | 1.00 | 38.8 | 1.00 | |
| Theater awards[2] | Yes | 89 | 69 | 56.3 | 1.53(1.21–1.93) | 54.1 | 1.44(1.22–1.70) | <0.001 |
| | No | 407 | 699 | 36.8 | 1.00 | 37.6 | 1.00 | |
| 5 MGEs attended[2] | | 6 | 1 | 85.7 | 4.97(2.16–11.47) | 77.7 | 4.11(3.25–5.19) | <0.001 |
| 4 MGEs attended[2] | | 52 | 18 | 74.3 | 4.31(3.00–6.20) | 75.9 | 4.01(3.08–5.23) | <0.001 |
| 3 MGEs attended[2] | | 95 | 53 | 64.2 | 3.72(2.72–5.10) | 61.2 | 3.24(2.49–4.21) | <0.001 |
| 2 MGEs attended[2] | | 125 | 116 | 51.9 | 3.01(2.23–4.06) | 49.4 | 2.61(2.02–3.39) | <0.001 |
| 1 MGE attended[2] | | 152 | 263 | 36.6 | 2.13(1.59–2.84) | 36.5 | 1.93(1.49–2.49) | <0.001 |

(*Continued*)

**Table 4.** (Continued)

| Variables | Cases | Non-cases | AR (%) | RR (95% IC) | aAR[1] (%) | aRR (95% IC)[1] | p-value |
|---|---|---|---|---|---|---|---|
| 0 MGEs attended[2] | 66 | 317 | 17.2 | 1.00 | 18.9 | 1.00 | |

1. Adjusted for age, sex, chronic illness, social class, family COVID-19 case, *falla*, and MGEs attended.

2. Adjusted for all factors except MGEs.

3. Age 15 years and above.

2.15 (95% CI 1.80–2.56), followed by the Queen's gala dinner with aRR of 1.90 (95% CI 1.67–2.17). The other MGEs had lower aRR, but still presented a significant and considerable risk (Table 4). When MGE exposure was measured by the number of events the participants had attended, a dose-response relationship was found with raw and adjusted AR. The aAR for participants who attended no MGEs was 18.9%, increasing to 77.7% for participants who attended five MGEs (aRR = 4.11 95% CI 3.25–5.19).

**Analysis of the *pa-i-porta* event.** The analysis of the MGE with the highest number of participants, the *pa-i-porta*, together with different exposure variables, is shown in Table 5.

The aRR by sex was similar in females and males. Interestingly, COVID-19 disease was marginally associated with the total time attended in a dose-response relationship from less than half of the event up to the whole event (aRR = 1.32 95% CI 0.98–1.74). The food or alcoholic beverages provided by the event organizers, or those brought to the event by the participants (which included cakes or other foods) were not associated with the disease. Participants who did not participate in the dinner and only attended the dance had a lower risk, but with limited effect (aRR = 0.84 95% CI 0.67–1.16). In addition, more time spent dancing was a marginal risk factor (aRR = 1.19 95% CI 0.96–1.48). Two groups of participants were defined according to whether or not they shared a dinner table with a participant reporting onset of COVID-19 symptoms between February 26 and March 6. The exposed group presented a higher aAR of disease than those who were not exposed, 55.7% versus 25.5% (aRR = 2.19 95% CI 1.40–3.42). Finally, the location and distribution of the dinner tables in building A were analyzed. To this end, the dinner tables were classified into four quadrants and an adjusted analysis was carried out. Results showed that the upper right quadrant had a higher COVID-19 incidence than the lower right quadrant (aRR = 1.44 95% 1.12–1.85), but no difference was found with the other quadrants (Table 5).

**Analysis of the Queen's gala dinner event.** The Queen's gala dinner was the MGE with the second highest number of participants in the study (Table 6). The aRR by sex was similar in females and males. The time of attendance at this event presented a dose-response relationship from less than half attendance to whole event attendance (aRR = 1.76 95% CI 1.18–2.62). At dinner, several dishes were consumed: starters (tomato bread, pork sausages, cheese and *foie gras* salad and scallops), second course (sirloin), and dessert (chocolate cream). The consumption of these foods and alcoholic beverages was not associated with COVID-19 risk. Next, two groups of participants were defined according to whether or not they shared a dining table with a participant with COVID-19 disease and symptoms onset between February 27 and March 7. The exposed group showed a higher aAR than the non-exposed group (76.1% and 58.6%, respectively) (aRR = 1.30 95% 1.10–1.53). The adjusted analysis of the incidence of COVID-19 cases across the four quadrants of tables in building A (same building as for the *pa-i-porta* event), revealed that the lower left quadrant had the highest COVID-19 incidence and differed from that of the lower right quadrant, which was the one with the lowest incidence (aRR = 1.29 95% 1.00–1.65). No differences were found with the other table quadrants

**Table 5. Pa-i-porta, Mass Gathering Event (MGE) attendance, Attack Rate (AR), adjusted AR (aAR), Relative Risk (RR), and adjusted (aRR).** Confidence interval (CI). From March 6 to 31, 2020, Borriana.

| Variables | Cases | Non- cases | AR (%) | RR (95% IC) | aAR (%) | aRR(95% IC)[1] | p-value |
|---|---|---|---|---|---|---|---|
| Total | 386 | 367 | 51.3 | | | | |
| Male | 143 | 132 | 52.0 | 1.02(0.83–1.26) | 49.3 | 0.96(0.83–1.11) | 0.407 |
| Female | 243 | 235 | 50.8 | 1.00 | 51.5 | | |
| Attendance time[2] | | | | | | | |
| Less than half | 32 | 37 | 46.4 | 1.00 | 44.2 | 1.0 | |
| Half | 111 | 143 | 43.7 | 0.94(0.64–1.40) | 44.5 | 1.01(0.75–1.35) | 0.956 |
| More than half | 104 | 86 | 54.7 | 1.18(0.80–1.75) | 53.6 | 1.21(0.91–1.62) | 0.189 |
| All the time | 138 | 100 | 58.0 | 1.25(0.85–1.84) | 58.4 | 1.32(0.98–1.74) | 0.051 |
| Dinner[2] | | | | | | | |
| Ate own food | 67 | 63 | 51.5 | 0.97(0.74–1.27) | 55.5 | 1.07(0.88–1.29) | 0.505 |
| Ate *falla* food | 268 | 238 | 53.0 | 1.00 | 52.0 | 1.00 | |
| Ate other food[2] | | | | | | | |
| Yes | 110 | 92 | 54.5 | 1.08(0.86–1.36) | 51.0 | 1.00(0.85–1.18) | 0.839 |
| No | 225 | 222 | 50.3 | 1.00 | 50.8 | 1.00 | |
| Cake consumption[2] | | | | | | | |
| Yes | 218 | 182 | 54.5 | 1.14 (0.99–1.43) | 53.2 | 1.10(0.94–1.28) | 0.236 |
| No | 120 | 132 | 47.6 | 1.00 | 48.7 | 1.00 | |
| Drank alcohol beverages at this event[2 3] | | | | | | | |
| Yes | 251 | 217 | 53.6 | 0.95(0.74–1.21) | 54.0 | 0.98(0.83–1.16) | 0.913 |
| No | 87 | 67 | 56.5 | 1.00 | 54.9 | 1.00 | |
| Attendance[2] | | | | | | | |
| Only dance | 43 | 58 | 42.6 | 0.81(0.59–1.11) | 44.2 | 0.84 (0.67–1.06) | 0.137 |
| All event | 332 | 300 | 52.5 | 1.00 | 52.5 | 1.00 | |
| Dance[2] | | | | | | | |
| Not at all | 94 | 93 | 51.3 | 1.00 | 47.6 | 1.00 | |
| A little | 104 | 100 | 51.0 | 1.01(0.77–1.34) | 48.5 | 1.04(0.84–1.29) | 0.708 |
| Some of the time | 103 | 97 | 51.5 | 1.02(0.77–1.35) | 53.2 | 1.12(0.91–1.38) | 0.292 |
| A lot | 82 | 71 | 53.6 | 1.06(0.79–1.43) | 56.8 | 1.19(0.96–1.48) | 0.120 |
| Dinner table with a COVID-19 symptomatic case[4] | | | | | | | |
| Yes | 338 | 311 | 52.4 | 1.18(0.86–1.62) | 55.7 | 2.19(1.40–3.42) | 0.001 |
| No | 44 | 56 | 44.0 | 1.00 | 25.5 | | |
| Quadrant of building A[5] | | | | | | | |
| Upper left | 130 | 118 | 52.1 | 1.36 (1.01–1.84) | 47.0 | 1.10 (0.79–1.52) | 0.570 |
| Lower left | 118 | 113 | 51.1 | 1.32 (0.98–1.81) | 51.7 | 1.18 (0.82–1.16) | 0.309 |
| Upper right | 71 | 35 | 67.0 | 1.74 (1.24–2.45) | 67.7 | 1.44 (1.12–1.85) | 0.005 |
| Lower right | 63 | 101 | 38.4 | 1.00 | 40.0 | 1.00 | |

1. Adjusted for age, sex, chronic illness, social class, family COVID-19 case, and *falla*.

2. Missing information of two cases.

3. Age 15 years and above.

4. Adjusted age, sex, mean of attendance time, and mean MGEs attended.

5. Adjusted for median of MGEs attended.

(Table 6). Interestingly, the lower right quadrant was the one with the lowest COVID-19 incidence in both the *pa-i-porta* and the Queen's gala dinner events.

Finally, our analysis also found that four out of the five food handlers who served in the two MGE dinners were confirmed COVID-19 cases by laboratory tests. Three of them reported

having COVID-19 symptomatology starting on March 16, 17, and 24 March 2020, respectively, after the MGEs were over. One case was asymptomatic.

**Sensitivity analysis.** The sensitivity analysis included the 1064 participants with laboratory tests of COVID-19 and cases between 6 and 31 March 2020 (Tables 7 and 8). The AR of this group was 44.0% from 468 COVID-19 cases and 596 non-cases. Some variations with the previous findings were found, considering its lower sample size and the higher age of participants (Table 2). Risk factors such as age and BMI lost significance, but alcohol consumption was associated with COVID-19 disease. Other factors did not show considerable changes. The *pa-i-porta* and the Queen's gala dinner maintained significant associations with COVID-19,

**Table 6. Queen's gala dinner, Mass Gathering Event (MGE) attendance, Attack Rate (AR), adjusted AR (aAR), Relative Risk (RR), and adjusted (aRR).** Confidence interval (CI). From March 6 to 31, 2020, Borriana.

| Variable | Cases | Non-cases | AR (%) | RR (95% IC) | aAR (%) | aRR(95% IC)[1] | p-value |
|---|---|---|---|---|---|---|---|
| Total | 198 | 110 | 64.3 | | | | |
| Male | 75 | 44 | 63.0 | 0.97(0.72–1.29) | 62.1 | 0.96(0.80–1.15) | 0.653 |
| Female | 123 | 66 | 65.1 | 1.00 | 64.7 | 1.00 | |
| Attendance time | | | | | | | |
| Less than half | 11 | 13 | 45.8 | 1.00 | 39.8 | 1.00 | |
| Half | 31 | 16 | 66.0 | 1.44 (0.78–2.88) | 65.3 | 1.64(1.05–2.55) | 0.028 |
| More than half | 42 | 31 | 57.5 | 1.26(0.64–2.44) | 56.3 | 1.42(0.91–2.20) | 0.121 |
| All time | 114 | 50 | 69.5 | 1.52(0,82.2.82) | 70.2 | 1.76(1.18–2.62) | 0.005 |
| Dinner | | | | | | | |
| Tomato bread Yes | 185 | 101 | 64.7 | 0.94(0.51–1.73) | 65.0 | 1.14(0.89–1.46) | 0.312 |
| No | 11 | 5 | 68.8 | 1.00 | 57.1 | 1.00 | |
| Pork sausages Yes | 182 | 99 | 64.8 | 0.97(0.56–1.67) | 64.7 | 0.88(0.71–1.09) | 0.249 |
| No | 14 | 7 | 66.7 | | 73.6 | 1.00 | |
| Foie salad Yes | 163 | 93 | 63.7 | 0.89(0.61–1.29) | 63.9 | 0.82(0.68–0.97) | 0.022 |
| No | 33 | 13 | 71.7 | | 78.4 | 1.00 | |
| Scallops Yes | 158 | 86 | 64.8 | 0.99(0.69–1.41) | 65.3 | 0.96(0.79–1.17) | 0.673 |
| No | 38 | 20 | 63.8 | | 68.1 | 1.00 | |
| Sirloin Yes | 173 | 96 | 64.3 | 0.92(0.60–1.43) | 65.0 | 0.77(0.67–0.89) | <0.001 |
| No | 23 | 10 | 69.7 | 1.00 | 84.4 | 1.00 | |
| Chocolate Yes | 165 | 82 | 66.8 | 1.19(0.81–1.74) | 67.5 | 1.13(0.91–1.40) | 0.263 |
| No | 31 | 24 | 54.5 | | 59.8 | 1.00 | |
| Drank alcohol [2] Yes | 163 | 94 | 63.4 | 0.93(0.65–1.35) | 63.6 | 1.20(0.95–1.52) | 0.123 |
| No | 34 | 14 | 70.8 | | 52.8 | 1.00 | |
| Dinner table with a COVID-19 symptomatic case[3] | | | | | | | |
| Yes | 88 | 35 | 71.5 | 1.18(0.89–1.57) | 76.1 | 1.30(1.10–1.53) | 0.002 |
| No | 103 | 67 | 60.6 | 1.00 | 58.6 | 1.00 | |
| Quadrant of building A[4] | | | | | | | |
| Upper left | 44 | 18 | 71.0 | 1.27 (0.84–1.91) | 63.4 | 0.98(0.72–1.34) | 0.917 |
| Lower left | 43 | 15 | 74.1 | 1.33 (0.88–2.00) | 83.0 | 1.29(1.00–1.65) | 0.047 |
| Upper right | 57 | 32 | 64.0 | 1.44 (0.78–1.68) | 64.6 | 1.00(0.76–1.32) | 0.990 |
| Lower right | 47 | 37 | 56.0 | 1.00 | 64.4 | 1.00 | |

1. Adjusted for age, sex, chronic illness, social class, family COVID-19 case, and *falla*.

2. Age 15 years and above.

3. Adjusted for age, sex, chronic illness, social class, family COVID-19 case, *falla*, attendance time, quadrant, and number of MGEs attended.

4. Adjusted for age, sex, chronic illness, social class, family COVID-19 case, *falla*, symptomatic case in dinner table, and number of MGEs attended.

**Table 7. Sensitivity analysis.** Participants with laboratory tests of SARS-CoV-2 and mass gathering event (MGEs) attendance, attack rate (AR) adjusted AR (aAR), and adjusted (aRR). Confidence interval (CI). From March 6 to 31, 2020, Borriana.

| Variables | Cases | Non-cases | AR (%) | aAR (%) | aRR (95% IC)[1] | p-value |
|---|---|---|---|---|---|---|
| Male | 180 | 249 | 42.0 | 44.1 | 1.01(0.89–1.16) | 0.862 |
| Female | 288 | 347 | 45.4 | 43.6 | 1.00 | |
| Age 0–4 (years)[2] | 4 | 14 | 22.2 | 16.4 | 1.00 | |
| 5–14 | 44 | 84 | 34.4 | 29.4 | 1.79(0.50–6.38) | 0.370 |
| 15–24 | 88 | 115 | 43.4 | 42.1 | 2.46(0.69–9.00) | 0.143 |
| 25–34 | 60 | 90 | 40.0 | 42.3 | 2.58(0.73–9.08) | 0.142 |
| 35–44 | 100 | 105 | 48.8 | 49.9 | 3.03(0.87–10.63) | 0.083 |
| 45–64 | 155 | 167 | 48.1 | 46.7 | 2.84(0.81–9.94) | 0.102 |
| 65 and above | 17 | 21 | 44.7 | 41.1 | 2.50(0.71–8.83) | 0.154 |
| Occupation I-II[1] | 142 | 122 | 53.8 | 51.6 | 1.23(1.07–1.42) | 0.003 |
| Occupation III-VI | 323 | 469 | 40.8 | 41.9 | 1.00 | |
| Chronic illness[1]      Yes | 155 | 184 | 42.9 | 43.7 | 0.99(0.87–1.15) | 0.992 |
| No | 309 | 411 | 45.7 | 43.9 | 1.00 | |
| Family with COVID-19    Yes | 293 | 213 | 57.9 | 55.6 | 1.64(1.43–1.88) | <0.001 |
| No | 174 | 376 | 31.6 | 33.9 | 1.00 | |
| Body Mass Index[1,3] <18.5 Kg/m$^2$ | 11 | 24 | 31.4 | 6.7 | 1.00 | |
| 18.5–24.9 Kg/m$^2$ | 189 | 252 | 42.9 | 45.6 | 6.81(0.41–113.91) | 0.182 |
| 25.0–29.9 Kg/m$^2$ | 139 | 154 | 47.4 | 48.8 | 7.29(0.44–121.98) | 0.167 |
| ≥30.0 Kg/m$^2$ | 79 | 66 | 54.5 | 59.1 | 8.83(0.53–401.56) | 0.130 |
| Physical exercise[1,3]      Yes | 244 | 322 | 43.1 | 44.1 | 0.90(0.78–1.02) | 0.109 |
| No | 176 | 176 | 50.0 | 49.2 | 1.00 | |
| Drank alcohol [1,3]      Yes | 104 | 108 | 49.1 | 53.4 | 1.19(1.04–1.37) | 0.015 |
| No | 316 | 390 | 44.8 | 44.8 | 1.00 | |
| Current smoker[1,3] | 68 | 169 | 28.7 | 30.7 | 0.62(0.50–0.75) | <0.001 |
| Ex smoker[1,3] | 104 | 83 | 55.6 | 54.8 | 1.10 (0.95–1.28) | 0.215 |
| Never smoker[1,3] | 246 | 245 | 50.1 | 49.8 | 1.00 | |
| 5 MGEs attended[2] | 6 | 1 | 85.7 | 82.6 | 3.62(2.83–4.61) | <0.001 |
| 4 MGEs attended[2] | 52 | 15 | 77.6 | 78.7 | 3.44(2.63–4.49) | <0.001 |
| 3 MGEs attended[2] | 94 | 48 | 66.2 | 64.3 | 2.81(2.15–3.67) | <0.001 |
| 2 MGEs attended[2] | 121 | 101 | 54.5 | 53.0 | 2.32(1.78–3.02) | <0.001 |
| 1 MGE attended[2] | 139 | 224 | 38.3 | 37.9 | 1.66(1.27–2.16) | <0.001 |
| 0 MGEs attended[2] | 56 | 207 | 21.3 | 22.9 | 1.00 | |

1. Adjusted for age, sex, chronic illness, social class, family COVID-19 case, *falla*, and MGEs attended.

2. Adjusted for all factors except MGEs.

3. Age 15 years and above.

including attendance time, table shared with a COVID-19 symptomatic participant, and quadrants in building A. In the Queen's gala dinner, quadrants lost significance.

## Discussion

The results of this study indicate a high transmission of COVID-19 disease in the MGEs studied, which may be considered as a community outbreak with several point sources from March 6 to 10 [47]. Respiratory transmission was the predominant mode of propagation by droplets from patients to exposed persons; contact via fomites was less frequent [48, 49]. Airborne transmission has been suggested in relation to COVID-19 outbreaks in different places,

**Table 8. Sensitivity analysis.** Participants with laboratory tests of SARS-CoV-2 and the pa-i-porta and Queen's gala dinner mass gathering event (MGE) attendance, attack rate (AR), adjusted AR (aAR), and adjusted (aRR). Confidence interval (CI). From March 6 to 31, 2020, Borriana.

| Variables | | Cases | Non-cases | AR (%) | aAR (%) | aRR (95%IC)[1] | p-value |
|---|---|---|---|---|---|---|---|
| *Pa-i-porta* | | | | | | | |
| Attendance[1] | Yes | 371 | 316 | 54.0 | 53.0 | 1.96(1.64–2.35) | <0.001 |
| | No | 97 | 280 | 25.7 | 27.0 | 1.00 | |
| Attendance time[1] [2] | | | | | | | |
| Less than half | | 30 | 33 | 47.6 | 44.6 | 1.00 | |
| Half | | 102 | 119 | 46.2 | 45.6 | 1.02(0.75–1.38) | 0.887 |
| More than half | | 102 | 74 | 58.0 | 52.3 | 1.26(0.94–1.70) | 0.122 |
| All the time | | 136 | 89 | 60.4 | 61.2 | 1.37(1.03–1.83) | 0.031 |
| Dinner table with a COVID-19 symptomatic case[3] | | | | | | | |
| | Yes | 326 | 263 | 55.3 | 58.8 | 2.57(1.45–4.83) | 0.001 |
| | No | 41 | 46 | 47.1 | 22.9 | 1.00 | |
| Quadrants building A[4] | | | | | | | |
| Upper left | | 127 | 93 | 57.7 | 51.8 | 1.25(0.88–1.77) | 0.212 |
| Lower left | | 112 | 98 | 53.3 | 53.3 | 1.29(0.95–1.74) | 0.105 |
| Upper right | | 69 | 31 | 69.0 | 69.3 | 1.67(1.34–2.08) | 0.000 |
| Lower right | | 59 | 87 | 40.4 | 41.5 | 1.00 | |
| Queen's gala dinner | | | | | | | |
| Attendance[1] | Yes | 196 | 98 | 66.7 | 65.5 | 1.80(1.58–2.05) | <0.001 |
| | No | 272 | 498 | 35.3 | 36.4 | 1.00 | |
| Attendance time[1] | | | | | | | |
| Less than half | | 11 | 11 | 50.0 | 42.4 | 1.00 | |
| Half | | 31 | 14 | 68.9 | 69.9 | 1.65(1.08–2.51) | 0.019 |
| More than half | | 42 | 26 | 61.8 | 61.0 | 1.44(0.94–2.18) | 0.086 |
| All the time | | 112 | 47 | 70.4 | 71.0 | 1.67 (1.15–2.45) | 0.008 |
| Dinner table with a COVID-19 symptomatic case[5] | | | | | | | |
| | Yes | 88 | 31 | 74.0 | 74.7 | 1.19 (1.01–1.40) | 0.037 |
| | No | 101 | 61 | 62.4 | 62.8 | 1.00 | |
| Quadrant of building A[6] | | | | | | | |
| Upper left | | 43 | 18 | 70.5 | 66.9 | 1.01(0.73–1.35) | 0.971 |
| Lower left | | 43 | 14 | 75.4 | 84.9 | 1.26(0.98–1.63) | 0.075 |
| Upper right | | 56 | 26 | 68.3 | 63.8 | 0.95(0.71–1.27) | 0.718 |
| Lower right | | 47 | 34 | 58.0 | 67.2 | 1.00 | |

1. Adjusted for age, sex, chronic illness, social class, family COVID-19 case, and *falla*.

2. Missing information in two cases.

3. Adjusted age, sex, median of attendance time, and median MGEs attended.

4. Adjusted for median of MGEs attended.

5. Adjusted for age, sex, chronic illness, social class, family COVID-19 case, *falla*, attendance time, quadrant, and number of MGEs attended.

6. Adjusted for age, sex, chronic illness, social class, family COVID-19 case, *falla*, symptomatic case in dinner table, and number of MGEs attended.

and is a plausible route [50–52]. The two MGEs with the highest incidence of the disease included a dinner, as in other COVID-19 outbreaks, but food-borne transmission can be discarded in this study [53, 54].

Some characteristics of this MEG outbreak may be highlighted, including its magnitude and impact in the population of Borriana, the diversity of places where it occurred indoors (building and theater) and outdoors (square), the higher risk of COVID-19 associated with the number of MGEs attended, the estimation of risk for attendance at no MGEs, the detection of

risk factors (obesity, old age, and upper and middle class versus lower class) with current smoking as a protective factor, and the rapid spread with high ARs.

In the study, the AR was 42.6%, and during the period of the MGEs, 39.2% with 94.0% of confirmed cases. If the AR were extrapolated to *falles* members, the total of COVID-19 cases could be between 1193 (95% CI 1269–1117) and 1098 (95% CI 1173–1023), respectively. In addition, a high proportion of cases had family members with COVID-19, suggesting a high secondary transmission among families, in line with Thompson and co-authors and with a study of secondary attack rate of COVID-19 infections in Castellon [55, 56]. It may explain the high incidence in the municipality of Borriana, which was the highest in the province of Castellon during the first outbreak period. Considering the period February to July 2020, the incidence of COVID-19 in Borriana was 307 cases with positive PCR results (885 per 100,000 inhabitants) and 40 deaths (1.2 per 1000 inhabitants) [34]. As a comparison, the COVID-19 incidence in Castellon de la Plana, the provincial capital (170,000 inhabitants), was 463 cases (272.4 per 100,000 inhabitants) and 50 deaths (0.29 per 1,000 inhabitants). The high mortality in Borriana is striking [57, 58], and it may be related to the prevalence of cardiovascular risk factors in this municipality [59]. Regarding the incidence of COVID-19, the study found 536 confirmed cases, 39 cases with positive PCR results; these were the only cases reported by the health authorities. It may be estimated that for every reported case, there could have been around 13 or 14 cases, implying that 92.7% of cases were undiagnosed. This situation is in line with two studies on undocumented infections of SARS-CoV-2 in China (86% and 93%), and France (86%), respectively [60–62].

In the context of COVID-19 outbreaks related to MGEs, different models have been proposed to estimate the number of cases deriving from MGEs or Ro values [63, 64]. Apart from assuming several premises, these models need a constant infection rate or Ro. As a tentative approximation and using the classic SIR (susceptible, infected, recovered) infection process model [65], we explored our results to obtain the Ro from the infection rate based on the date of onset of cases, number of contacts, and duration of infectiousness. We used the formula [66]:

$$Ro = PI \times CR \times DI$$

where PI = "average probability a contact will be infected over duration of a relationship"; CR = "average rate of getting into contact"; and DI = "average duration infectiousness". We defined PI, the infection rate, as new cases of COVID-19 divided by the susceptible participants per day; CR as a number of potential contacts, following the study of Tupper and co-authors [63], and DI as 10 days duration of infectiousness [67]. We considered a period of 19 days from the first MGE to the last MGE, plus a 14-day maximum incubation period for COVID-19. The results are shown in Table 9.

Considering the infection rate per day, an increase was observed from 0.00172 on the first day to 0.04810 on March 16 and a decline until 0.00811 on March 25. The countrywide lockdown began on March 15 and has been shown to have had a high effect [68]. With five contacts, Ro>1.0 reached only 8 days and a maximum of 2.41; with 10 contacts, Ro>1.0 reached 13 days and a maximum of 4.81; and with 15 contacts, Ro>1.0 reached 19 days and a maximum of 7.22 on the 16th day. The decline of the Ro was very steep from the maximum and Ro<1.0 values were observed, indicating that the epidemic would soon decline [69]. These results reflect the high transmission of the disease associated with MGEs.

To explore the expected numbers of cases in the COVID-19 outbreak associated with the MGEs, we followed the approach of Tupper and co-authors [63], considering the probability of infection (β), the duration time (T), time of contact (τ) and the number of contacts (k). We

**Table 9. Estimation of the basic reproductive number (Ro) from the MGE COVID-19 outbreak between 6 and 10 March 2020 plus 14 days.**

| Days/Date infection onset | New cases (a) | Total cases (I) (b) | Susceptible (S) (c) | Infection rate (a/c) | Basic reproduction number Ro[1] Numbers of contacts[2] | | |
|---|---|---|---|---|---|---|---|
| | | | | | 5 | 10 | 15 |
| 3/6/2020: 0[3] | 9 | 9 | 1264 | 0.00712 | 0.36 | 0.71 | 1.07 |
| 3/7/2020: 1[3] | 11 | 20 | 1255 | 0.00877 | 0.44 | 0.88 | 1.32 |
| 3/8/2020: 2[3] | 15 | 35 | 1244 | 0.01206 | 0.60 | 1.21 | 1.81 |
| 3/9/2020: 3 | 45 | 80 | 1229 | 0.03662 | 1.83 | 3.66 | 5.49 |
| 3/10/2020[3] 4 | 39 | 119 | 1184 | 0.03294 | 1.65 | 3.29 | 4.94 |
| 3/11/2020: 5 | 32 | 151 | 1145 | 0.02795 | 1.40 | 2.80 | 4.19 |
| 3/12/2020: 6 | 21 | 172 | 1113 | 0.01887 | 0.94 | 1.89 | 2.83 |
| 3/13/2020: 7 | 31 | 203 | 1092 | 0.02839 | 1.42 | 1.84 | 4.26 |
| 3/14/2020: 8 | 33 | 236 | 1061 | 0.03110 | 1.56 | 3.11 | 4.67 |
| 3/15/2020: 9 | 30 | 266 | 1028 | 0.02918 | 1.46 | 2.92 | 4.38 |
| 3/16/2020:10 | 48 | 314 | 998 | 0.04810 | 2.41 | 4.81 | 7.22 |
| 3/17/2020:11 | 20 | 334 | 950 | 0.02105 | 1.03 | 2.05 | 3.08 |
| 3/18/2020:12 | 18 | 352 | 930 | 0.01935 | 0.97 | 1.94 | 2.90 |
| 3/19/2020:13 | 15 | 369 | 912 | 0.01644 | 0.82 | 1,64 | 2.47 |
| 3/20/2020:14 | 15 | 382 | 897 | 0.01672 | 0.84 | 1.67 | 2.51 |
| 3/21/2020:15 | 6 | 388 | 882 | 0.00680 | 0.34 | 0.68 | 1.02 |
| 3/22/2020:16 | 7 | 395 | 876 | 0.00799 | 0.40 | 0.80 | 1.20 |
| 3/23/2020:17 | 2 | 397 | 869 | 0.00230 | 0.12 | 0.23 | 0.35 |
| 3/24/2020:18 | 4 | 401 | 867 | 0.00461 | 0.23 | 0.46 | 0.69 |
| 3/25/2020:19 | 7 | 408 | 863 | 0.00811 | 0.41 | 0.81 | 1.22 |

1. Ro = infection rate x contact numbers x duration infectiousness (10 days).

2. Considering average numbers of contacts in conferences, lectures, and restaurants (5, 10, and 15).

3. Date with MGE.

calculated k from two MGEs (*pa-i-porta* and the Queen's gala dinner). The formula yields the expected number of new infections:

$$R_{event} = (kT/\tau)(1 - e^{\beta\tau})$$

Attendances at the *pa-i-porta* and Queen's gala dinner events were used to estimate expected cases and compare them with observed cases. For the *pa-i-porta* event, the number of contacts was 85 for less than half the attendance time and 60 for more than half the attendance time, to obtain a similar number of observed and expected cases. For the Queen's gala dinner the numbers of contacts were 20 for less than half the attendance time and 40 for more than half the attendance time. Considering that 9 and 11 cases of COVID-19 (Table 10) had the onset of symptoms the day of the *pa-i-porta* and the Queen's gala dinner, respectively, the number of contacts k could be divided by 9 and 11, and contacts could be 7 to 10 for *pa-i-porta* and 2 to 4 for the Queen's gala dinner. These results could explain the spread of the epidemic, and underline the importance of the duration of the event, the number of contacts, and the number of participants, which were much higher in the *pa-i-porta* than in the Queen's gala dinner. However, a major limitation of this approach is the theoretical assumption of homogeneous mixing of population in the epidemic [70]. In addition, asymptomatic or undiagnosed cases may bias the results and 67 cases were asymptomatic in our study.

**Table 10. Attendance at *pa-i-porta* and Queen's gala dinner MGEs and number of contacts (k) to obtain expected cases of COVID-19 compared with observed cases from the formula of Tupper and co-authors [63].**

| MGEs | Attendance time | Probability infection β | Time T hours | Contact time τ | Number contacts k | Number expected Cases | Number observed cases |
|------|-----------------|-------------------------|--------------|----------------|-------------------|-----------------------|-----------------------|
| *Pa-i-porta* | Less than half | 0.465 | 3.30 | 0.2 | 85 | 132.1 | 132 |
| | More than half | 0.594 | 7.0 | 0.2 | 60 | 235.2 | 238 |
| Queen's gala dinner | Less than half | 0.627 | 3.15 | 0.2 | 20 | 41.2 | 42 |
| | More than half | 0.678 | 6.30 | 0.2 | 40 | 152.2 | 154 |

1. Infection rates of attendance at *pa-i-porta* and Queen's gala dinner
2. Contact time from Tupper and co-authors [63].

In our study the iceberg-like pattern of COVID-19 could be observed, considering the percentages of 0.17% patient deaths, 2.3% hospitalized, 12% asymptomatic, and 85.8% symptomatic [71]. The most frequent symptoms of COVID-19 cases were weakness, fever, and lost smell or/and taste; the severe course of the disease was less frequent in accordance with the average age of the patients [59, 72, 73]. In addition, the medical assistance was low and chronic illness was weakly associated with COVID-19, but cardiovascular diseases, hypertension, and hypothyroidism increased the risk of COVID-19, as indicated in some studies [74, 75]. The presence of sequels remains low, but this is a new disease and follow-up of these patients would be useful.

Considering the risk factors of COVID-19, higher age was associated with the disease, and children and adolescents were less affected [76, 77]; in contrast with other studies [78], sex was not a risk factor. Young patients constituted asymptomatic cases, and their frequency was lower than in other studies [79–81]. The validity of self-reported lifestyle data has been considered good [82], and overweight and obesity were risk factors in line with population-based studies of COVID-19 [83, 84]. In addition, Merzon and co-authors [85] found that vitamin D deficiency was a risk factor of COVID-19 infection and severity. Alcohol consumption was a marginal risk factor, but has not been found in some studies [86, 87]. Upper and middle social classes were associated with the disease, in contrast with other studies [85, 88]. Smoking is a controversial factor because it has been found to be both protective and a risk factor [85–87, 89, 90].

When observing the MGEs, it may be useful to consider that participation in the *Falles* festival was extensive, since the population of Borriana, both adults and children, took part in many cultural, touristic, leisure, and dinner events [35], and many MGEs were held in February and March 2020. These MGEs may be considered as super-spreading events [91, 92]. In this context, several points stand out. First, COVID-19 cases began in January and February 2020, with the onset of symptoms before the MGEs took place. Second, the *pa-i-porta* and the Queen's gala dinner included dancing, which involved more contact among participants. In these MGEs the presence of participants with symptoms at the same dinner table was a significant risk factor of COVID-19, which is consistent with symptomatic cases being more contagious than asymptomatic cases [10, 93, 94]. Third, indoor MGEs had a high attendance, mainly in building A; considering the building's capacity and the close proximity among participants, a higher risk of transmission was found [3, 95, 96]. Attendance at the other MGEs was lower and full capacity was not reached. Fourth, several MGEs lasted for more than six hours and ended in the early morning. The average temperature in Borriana in March is 13.4˚C, and the average relative humidity is 64% [97]. These weather conditions have been shown to favor SARS-CoV-2 transmission, estimated to range between 5˚-15°C and 30–100% of temperature and relative humidity, respectively [98, 99]. Fifth, all the MGEs except the Valencia trip occurred in closed

indoor places, building A and theater B; both locations have air conditioning and heating, which are more modern in the case of theater B, and no breakdowns or malfunctions were observed. Finally, airborne transmission [100] may be possible, and our results indicate one quadrant of building A with a lower incidence of COVID-19, but the two quadrants with the highest incidence were not the same in the two events, and these two quadrants were occupied by the *Falles* organizers and guests, who had attended the most MGEs. However, the presence of asymptomatic cases, the variability of the incubation period, and other potential transmission sources make it difficult to confirm this transmission in these MGEs.

A potential causal relationship between MGE attendance and COVID-19 may be considered following the Austin Bradford-Hill criteria [101]; aRR of MGE attendance ranged from moderate to high. It has been suggested that there is a biological plausibility of COVID-19 transmission happening in closed places, where many participants gather in areas that are smaller than 1.83 m$^2$ per individual [102]. This is consistent with other COVID-19 outbreaks in MGEs in several countries [21, 24, 42, 103, 104]. The temporal relationship between MGE attendance and COVID-19 was established after the event. A dose-response relationship was demonstrated between MGE attendance, attendance time, and risk of the disease [105]. As an observational study, a retrospective cohort design could estimate the causality relationship more precisely than descriptive studies.

The strengths of our study are as follows. First, it has a population-based design that allows a more integral approach to COVID-19. Second, a representative sample of a population exposed to MGEs was analyzed. Third, the response rate was high. Fourth, a serologic survey was carried out to confirm COVID-19 cases [106, 107] with a highly sensitive and specific technique. Fifth, statistical analysis was adjusted for potential confounding factors. Finally, the sensitivity analysis confirmed the previous results with an improvement in study precision.

The study's limitations are as follows. First, there was a period of time between exposure at the MGEs and the start of the study. In addition, the impact of the COVID-19 pandemic could have caused some recall and misclassification biases. Second, only the *Falles* population was included in the study. Third, other MGEs were not studied, such as the *Ninot* parade, community dinners, children's entertainments, and others that may have played a role in COVID-19 transmission in Borriana. Fourth, an estimation of incubation periods was difficult considering the variety of MGE attendance. Fifth, no genetic studies were carried out into the SARS-CoV-2 of positive PCR patients. Sixth, some residual biases may persist despite adjustments. Finally, as a new disease, COVID-19 could have some factors that were not considered in the study.

Some recommendations may be addressed based on the results of the study. First, and with immediate effect, MGEs should be considered as potential triggers of high transmission of SARS-CoV-2. The maintenance, conservation, and inspection of the closed indoor building where these MGEs took place should also be strengthened. We must highlight that MEGs have continued to be held successfully with measures in place to prevent SARS-CoV-2 transmission, suggesting that MGEs could be implemented again with specific prevention measures [108–112]. Second, COVID-19 patients should be monitored to find out about this new disease and its evolution. Third, population-based studies, including serologic surveys, should be carried out to estimate the extent of COVID-19 [113, 114]. Finally, regarding our study, new lines of research could be implemented, including estimating other risk factors of COVID-19 disease, determining serologic variations of antibodies against SARS-CoV-2 in the coming months, following-up patients and potential sequels, and exploring factors associated with the spread of the COVID-19 pandemic [115].

As an epilogue to our study, we are carrying out a prospective cohort study with patients who tested positive for COVID-19, and a second sero-epidemiological study, started in October 2020, to research evolution, sequelae and antibodies against SARS-CoV-2.

## Conclusions

The results of this study suggest the importance of MGEs in COVID-19 transmission that could explain the subsequent COVID-19 outbreak in Borriana. Population-based studies, including serologic COVID-19 surveys, may usefully inform the adoption of preventive measures that help to contain the COVID-19 pandemic.

## Supporting information

**S1 File.**
(DOCX)

**S1 Data.**
(DTA)

**S2 Data.**
(DTA)

**S1 Dataset.**
(DOCX)

## Acknowledgments

We are grateful to all the people who participated in the study and to the organizers of the *Falles* in Borriana for their help and support. In addition, we wish to thank Roser Blasco-Gari, Helena Buj-Jorda, Israel Borras-Acosta, Lucia Castell-Agusti, Mercedes De Francia-Valero, Maria Domènech-Molinos, Marc Garcia, Maria Gil-Fortuño, Elena Grañana-Toran, Noelia Hernández-Perez, Laura Lopez-Diago, Salvador Martinez-Parra, Sara Moner-Marin, Silvia Pesudo-Calatayud, Lara Sabater-Hernández, Maria Luisa Salve-Martinez, Irene Suarez-Linares, Juan José Ventura-Buchardo, and Alberto Yagüe-Muñoz for their assistance and support in carrying out the study. We acknowledge the help of Francis Conde-Mollà, Cristina Devis-Drago, and Carolina Vicente-Rodriguez in implementing this study, Jose Monserrat-Vicent for his assistance with the English and Mary B Savage Cooper for her English revision. We thank Claudia Arnedo-Pac and Jose Antonio Valer for their advice and for constructing the figures.

## Author Contributions

**Conceptualization:** Salvador Domènech-Montoliu, Maria Rosario Pac-Sa, Paula Vidal-Utrillas, Marta Latorre-Poveda, Alba Del Rio-González, Sara Ferrando-Rubert, Gema Ferrer-Abad, Manuel Sánchez-Urbano, Laura Aparisi-Esteve, Gema Badenes-Marques, Belén Cervera-Ferrer, Ursula Clerig-Arnau, Claudia Dols-Bernad, Maria Fontal-Carcel, Lorna Gomez-Lanas, David Jovani-Sales, Maria Carmen León-Domingo, Maria Dolores Llopico-Vilanova, Mercedes Moros-Blasco, Cristina Notari-Rodríguez, Raquel Ruíz-Puig, Sonia Valls-López, Alberto Arnedo-Pena.

**Data curation:** Salvador Domènech-Montoliu, Maria Rosario Pac-Sa, Paula Vidal-Utrillas, Marta Latorre-Poveda, Alba Del Rio-González, Sara Ferrando-Rubert, Gema Ferrer-Abad, Manuel Sánchez-Urbano, Laura Aparisi-Esteve, Gema Badenes-Marques, Belén Cervera-Ferrer, Ursula Clerig-Arnau, Claudia Dols-Bernad, Maria Fontal-Carcel, Lorna Gomez-Lanas, David Jovani-Sales, Maria Carmen León-Domingo, Maria Dolores Llopico-Vilanova, Mercedes Moros-Blasco, Cristina Notari-Rodríguez, Raquel Ruíz-Puig, Sonia Valls-López, Alberto Arnedo-Pena.

**Formal analysis:** Salvador Domènech-Montoliu, Maria Rosario Pac-Sa, Gema Ferrer-Abad, Alberto Arnedo-Pena.

**Investigation:** Salvador Domènech-Montoliu, Maria Rosario Pac-Sa, Paula Vidal-Utrillas, Marta Latorre-Poveda, Alba Del Rio-González, Sara Ferrando-Rubert, Gema Ferrer-Abad, Manuel Sánchez-Urbano, Laura Aparisi-Esteve, Gema Badenes-Marques, Belén Cervera-Ferrer, Ursula Clerig-Arnau, Claudia Dols-Bernad, Maria Fontal-Carcel, Lorna Gomez-Lanas, David Jovani-Sales, Maria Carmen León-Domingo, Maria Dolores Llopico-Vilanova, Mercedes Moros-Blasco, Cristina Notari-Rodríguez, Raquel Ruíz-Puig, Sonia Valls-López, Alberto Arnedo-Pena.

**Methodology:** Salvador Domènech-Montoliu, Maria Rosario Pac-Sa, Paula Vidal-Utrillas, Marta Latorre-Poveda, Alba Del Rio-González, Sara Ferrando-Rubert, Gema Ferrer-Abad, Manuel Sánchez-Urbano, Laura Aparisi-Esteve, Gema Badenes-Marques, Belén Cervera-Ferrer, Ursula Clerig-Arnau, Claudia Dols-Bernad, Maria Fontal-Carcel, Lorna Gomez-Lanas, David Jovani-Sales, Maria Carmen León-Domingo, Maria Dolores Llopico-Vilanova, Mercedes Moros-Blasco, Cristina Notari-Rodríguez, Alberto Arnedo-Pena.

**Project administration:** Salvador Domènech-Montoliu, Maria Rosario Pac-Sa.

**Resources:** Laura Aparisi-Esteve.

**Software:** Maria Rosario Pac-Sa, Alberto Arnedo-Pena.

**Supervision:** Salvador Domènech-Montoliu, Maria Rosario Pac-Sa, Paula Vidal-Utrillas, Gema Ferrer-Abad, Laura Aparisi-Esteve, Lorna Gomez-Lanas.

**Validation:** Maria Rosario Pac-Sa, Marta Latorre-Poveda, Manuel Sánchez-Urbano, Gema Badenes-Marques, Ursula Clerig-Arnau, Alberto Arnedo-Pena.

**Visualization:** Maria Rosario Pac-Sa, Paula Vidal-Utrillas, Alba Del Rio-González, Sara Ferrando-Rubert, Manuel Sánchez-Urbano, Ursula Clerig-Arnau, Claudia Dols-Bernad, Maria Fontal-Carcel, Lorna Gomez-Lanas, David Jovani-Sales, Maria Dolores Llopico-Vilanova, Mercedes Moros-Blasco, Cristina Notari-Rodríguez, Raquel Ruíz-Puig, Sonia Valls-López.

**Writing – original draft:** Alberto Arnedo-Pena.

**Writing – review & editing:** Salvador Domènech-Montoliu, Maria Rosario Pac-Sa, Paula Vidal-Utrillas, Marta Latorre-Poveda, Alba Del Rio-González, Sara Ferrando-Rubert, Gema Ferrer-Abad, Manuel Sánchez-Urbano, Laura Aparisi-Esteve, Gema Badenes-Marques, Belén Cervera-Ferrer, Ursula Clerig-Arnau, Claudia Dols-Bernad, Maria Fontal-Carcel, Lorna Gomez-Lanas, David Jovani-Sales, Maria Carmen León-Domingo, Maria Dolores Llopico-Vilanova, Mercedes Moros-Blasco, Cristina Notari-Rodríguez, Raquel Ruíz-Puig, Sonia Valls-López, Alberto Arnedo-Pena.

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
