## [Decision Letter · Decision Letter 0]

8 Apr 2021

PONE-D-21-04796

“MASS GATHERING EVENTS AND COVID-19 TRANSMISSION IN BORRIANA (SPAIN): A RETROSPECTIVE COHORT STUDY.”

PLOS ONE

Dear Dr. Arnedo-Pena

Thank you for submitting your manuscript to PLOS ONE. After careful consideration, we feel that it has merit but does not fully meet PLOS ONE’s publication criteria as it currently stands. Therefore, we invite you to submit a revised version of the manuscript that addresses the points raised during the review process.

I have read the comments provided by two expert reviewers and I read the mansucript myself too. I will not reiterate the Rs comments but I will add mine. I strongly recommend consulting PlosOne styling guide for submissions. At the moment, the ms looks like a draft: arbitary spaces, no line numbers, tables included in the main text, some of the tables are tilted, e.g. Table 7; as well as overal language. As Rs mention, there is merit in this study but it needs to substantial amount work on style.

We look forward to receiving your revised manuscript.

Kind regards,

Huseyin Cakal

Academic Editor

PLOS ONE

Journal Requirements:

Please correct your reference to "p=0.000" to "p<0.001" or as similarly appropriate, as p values cannot equal zero.

Please provide additional details regarding participant consent. In the ethics statement in the Methods and online submission information, please ensure that you have specified how verbal consent was documented and witnessed.

Please state in your methods section whether you obtained consent from parents or guardians of the minors included in the study or whether the research ethics committee or IRB approved the lack of parent or guardian consent.

Please include additional information regarding the survey or questionnaire used in the study and ensure that you have provided sufficient details that others could replicate the analyses. For instance, if you developed a questionnaire as part of this study and it is not under a copyright more restrictive than CC-BY, please include a copy, in both the original language and English, as Supporting Information. Moreover, please include more details on how the questionnaire was pre-tested, and whether it was validated.

In your Data Availability statement, you have not specified where the minimal data set underlying the results described in your manuscript can be found. PLOS defines a study's minimal data set as the underlying data used to reach the conclusions drawn in the manuscript and any additional data required to replicate the reported study findings in their entirety. All PLOS journals require that the minimal data set be made fully available. For more information about our data policy, please see http://journals.plos.org/plosone/s/data-availability.

Reviewers' comments:

Reviewer's Responses to Questions

**Comments to the Author**

1. Is the manuscript technically sound, and do the data support the conclusions?

Reviewer #1: Yes

Reviewer #2: Yes

2. Has the statistical analysis been performed appropriately and rigorously? 

Reviewer #1: Yes

Reviewer #2: N/A

3. Have the authors made all data underlying the findings in their manuscript fully available?

Reviewer #1: Yes

Reviewer #2: No

4. Is the manuscript presented in an intelligible fashion and written in standard English?

Reviewer #1: Yes

Reviewer #2: Yes

5. Review Comments to the Author

Reviewer #1: The manuscript entitled “MASS GATHERING EVENTS AND COVID-19 TRANSMISSION IN BORRIANA

(SPAIN): A RETROSPECTIVE COHORT STUDY” aims at estimating the incidence of COVID-19 disease, its association with MGEs and quantifying the potential risk factors of its occurrence.

The manuscript is interesting, nevertheless, it presents some flaws which make me conclude that before it is ready for publication in PLOS ONE the authors need to address some minor issues. Below are my comments to the authors and I hope these ideas will be helpful to them for improving their manuscript.

Abstract

The abstract seems clear, although I would recommend describing in more details the purpose of the study and the aims that are to be achieved. Additionally, it is important to highlight the novelty of the study.

Introduction

Although very short the introduction is also mostly clear and well written. However, I have some suggestions or inquire to clarify some minor issues:

1. The Introduction should present the literature to date and identify a gap in this research area, therefore I´m sure it would benefit from editing some of the content. Please, at least mention the definition of mass gatherings and some of its characteristics.

2. In the same vain, for instance, if the existing empirical evidence regarding the effects of mass gatherings provide ambiguous results, please explain what is unclear and why.

Method

The method section in exhaustive and well explained.

3. Would be helpful to have some more details regarding how the participants were recruited.

4.What other variables was used in the project, only those presented in the paper?

Results

The result section is well described and explained.

Discussion

5. Although the findings are interesting, and they do confirm the hypothesis I am not exactly sure how the current paper adds to what we already know. Therefore, it would be nice to introduce a short paragraph, which resumes the unique contributions of the study.

8. The writing is sometimes problematic and hard to understand and the flow a bit choppy. Therefore, I would recommend a thorough spell and grammar check of the entire paper.

12. In general, I think especially the discussion currently does not do a good job at all in highlighting the unique contributions of the paper.

Reviewer #2: The present research is devoted to analyze how mass gathering events linked to the “Falles” festival in Borriana are related to COVID-19 incidence, while also examining a number of relevant risk factors. The results highlight the importance of these mass gathering events in explaining COVID-19 incidence.

Although the article stated in the discussion section that studies estimating COVID-19 risk factors are limited, a broad range of them can be observed in the literature.

Overall, limitations of this research relate to the lack of a solid structure in the introduction section. This section only makes a brief reference to the general aim of this research.

Clear theoretical foundations and literature review closely related to topic under study are needed.

Description of MGEs during “Falles” festival in Borriana should be covered in further detail (as a complement, the creation of a summary table might be considered).

It is stated that the study was conducted from 14th May to 31st, June 2020, and that it comprises two phases. However, later on is mentioned that the fir phase started on April. In addition, at the same page it is stated that the study period ranged from January to June 2020. A more detailed, comprehensible, and homogenous description of the study’s design should be provided.

2.3% of participants were between the age range 0-4, and 14.2% between 5 and 14 (under legal minimum age in Spain). Could these participants adequately fill out the questionnaire survey? What procedure was followed? How certain sociodemographic variables (e.g., social class) were calculated on the basis of the abovementioned fact?

These aspects must be addressed as there is a considerable number of participants under 5 and 14 years old, thereby reflecting the need to provide more detailed description on procedure and methodological aspects.

Did the authors collect participants’ educational level? It should have been included as a control variable in the light of its connections with COVID-19 social behavior.

The inclusion of each potential risk factor (confounding factors) should be previously explained/justified.

The section Analysis of COVID-19 outbreak and MGEs should be divided in to “smaller” sections so as to facilitate a clear picture of the current results.

The discussion section should be addressed considering prior comments.

Certain grammatical and reading errors are still observed. This version must be proofread and edited by a native English-speaking expert.

6. PLOS authors have the option to publish the peer review history of their article (what does this mean?). If published, this will include your full peer review and any attached files.

Reviewer #1: **Yes: **Anna Wlodarczyk

Reviewer #2: No

---

## [Author Response · Author response to Decision Letter 0]

11 May 2021

Response to the Editor and Reviewers

PONE-D-21-04796

“MASS GATHERING EVENTS AND COVID-19 TRANSMISSION IN BORRIANA (SPAIN): A RETROSPECTIVE COHORT STUDY.”

PLOS ONE

Dear Mr. Huseyin Cakal

Thank you very much for your attention and help.

We have tried to follow all your indications and suggestions.

Journal Requirements:

1. The PLOS ONE style.

We have to try to follow the recommended PLOS OPEN style.

2. Please correct your reference to “p=0.000” to “p<0.001”

We have corrected the p values p=0.000 to p<0.001

3. Please provide additional details regarding participant consent.

We obtained a verbal consent of the participants before to perform the survey questionnaire. Several clarifications have made during the phone call as stated in the questionnaire.

-The participation in this study is voluntary, and in at any time you can leave the study, without damage on your part.

-You can receive information about the study.

-The personal information obtained from this study for the identification of each patient will be kept in the strictest confidentiality.

-The data collected will be used anonymously for scientific research, safeguarding your privacy.

Explicit consent: 

-Given the characteristics of the study and that the survey is carried out by telephone, it is understood that in addition to answering affirmatively to the question if you want to participate, the fact of answering the questionnaire ratifies said consent. 

-So that the research is carried out with scrupulous respect for ethical principles and there is no doubt about the voluntary participation, the researcher responsible for data collection is identified.

4. Please state in your methods section whether you obtained consent from parents of guardians of the minors.

When a child was chosen in the sampling, their parents were asked if they allowed their child to participate and answer the questionnaire with the help of the parents, considering the all clarifications before to perform the survey questionnaire. 

5. Please include additional information regarding the survey or questionnaire used in the study.

A copy of the questionnaire sample and its English translation are included in the supporting information. 

6. All PLOS journals require that the minimal data set be made fully available. Moreover, please include more details on how the questionnaire was pre-tested, and whether it was validated.

We add the Dataset of the study: the principal Dataset: borriana6661.dta and an additional Dataset:pa-i-porta6661.dta. The STATA® 14 version program was used.

Our questionnaire was not validated before. However, we used a questionnaire to study the secondary attack rate of COVID-19 infection in household contacts of Castellon, and the approach (telephone interview) and some questions were similar. The reference is 

Arnedo-Pena A, Sabater-Vidal S, Meseguer-Ferrer N, Pac-Sa R, Mañes-Flor P, Gascó-Laborda JC, et al. COVID-19 secondary attack rate and risk factors in household contacts in Castellon (Spain): Preliminary report. Rev Enf Emer 2020;19:64-70. 

Comments to the Author

Please use the space provided to explain your answers to the questions above. You may also include additional comments for the author, including concerns about dual publication, research ethics, or publication ethics. (Please upload your review as a

Reviewer #1: The manuscript entitled “MASS GATHERING EVENTS AND COVID-19 TRANSMISSION IN BORRIANA

(SPAIN):

Thank you very much for your indications and suggestions. We appreciate your positive attitude. 

-The abstract seems clear, although I would recommend describing in more details the purpose of the study and the aims that are to be achieved. Additionally, it is important to highlight the novelty of the study.

 We clarify the objective of the study. 

1. The Introduction should present the literature to date and identify a gap in this research area, therefore I´m sure it would benefit from editing some of the content. Please, at least mention the definition of mass gatherings and some of its characteristics. 

We have changed the introduction with more than 20 new references. We add the definition MGEs by the WHO, characteristics of MGEs, and several MGEs with COVID-19 transmission in different countries. References of control of MGEs by WHO and CDC are indicated and the basic reproductive number (Ro). . Some problems in the reporting of COVID-19 outbreaks and COVID-19 outbreaks in Spain are included in the introduction (lines 92-111).

2. In the same vain, for instance, if the existing empirical evidence regarding the effects of mass gatherings provide ambiguous results, please explain what is unclear and why.

Some biases in COVID-19 outbreak epidemiologic studies have been indicated, and we add references of this aspect (lines 112-116).

3. Would be helpful to have some more details regarding how the participants were recruited.

We explain in more detail how the participants were recruited means a numbered list of the members of each falla and the list of participants in the Queen gala dinner (lines 195-198).

4. What other variables was used in the project, only those presented in the paper? 

In order to try to test our hypothesis of the high incidence of COVID-19 disease, a longer questionnaire was used. Some variables like taking vitamins or medications, frequency of activities and sleeping during March 6-10, amount of food consumed during MGEs dinners, and means to travel to Valencia were not included in the paper. 

Discussion

5. Although the findings are interesting, and they do confirm the hypothesis I am not exactly sure how the current paper adds to what we already know. Therefore, it would be nice to introduce a short paragraph, which resumes the unique contributions of the study.

We add a paragraph to the summary of the study findings (lines 564-570).

8. The writing is sometimes problematic and hard to understand and the flow a bit choppy. Therefore, I would recommend a thorough spell and grammar check of the entire paper.

The final version of the manuscript has been reviewed by a native English-speaking professor.

12. In general, I think especially the discussion currently does not do a good job at all in highlighting the unique contributions of the paper.

We have made several changes in the discussion with new paragraph to summarize the findings of the study. In addition, we add two tables in order to apply the Ro to our data (lines 591-647). 

Table 9. Estimation of the basic reproductive number (Ro) from the MGE COVID-19 outbreak between 6 and 10 March 2020 plus 14 days

 Table 10. Attendance at pa-i-porta and Queen’s gala dinner MGEs and number of contacts (k) to obtain expected cases of COVID-19 compared with observed cases from the formula of Tupper and co-authors (63).

We have included new references of MEGs, how the MEGs are taking place now, and an epilog of our study: a cohort study of COVID-19 patients (lines 737-740). 

Reviewer #2: The present research is devoted to analyze how mass gathering events linked to the “Falles” festival in Borriana are related to COVID-19 incidence, while also examining a number of relevant risk factors. The results highlight the importance of these mass gathering events in explaining COVID-19 incidence.

Thank you very much for your indications and suggestions

Although the article stated in the discussion section that studies estimating COVID-19 risk factors are limited, a broad range of them can be observed in the literature. Overall, limitations of this research relate to the lack of a solid structure in the introduction section. This section only makes a brief reference to the general aim of this research. Clear theoretical foundations and literature review closely related to topic under study are needed.

We have improved the introduction section and have added a definition of MGEs from the WHO with MGEs studies of different countries, and have explained some limitations in the MGEs study such as the small size, few adjusted risks for confounders, and basic statistical analysis. In addition, we add references of COVID-19 outbreaks in Spain (lines 90-120).

Description of MGEs during “Falles” festival in Borriana should be covered in further detail (as a complement, the creation of a summary table might be considered).

We explain one by one the all MGEs and add a table with all these MGEs (lines 164-165) and Table 1.

Table 1. Characteristics of mass gathering events (MGEs) connected to the Falles festival in Borriana from March 6 to 10.

It is stated that the study was conducted from 14th May to 31st, June 2020, and that it comprises two phases. However, later on is mentioned that the fir phase started on April. In addition, at the same page it is stated that the study period ranged from January to June 2020. A more detailed, comprehensible, and homogenous description of the study’s design should be provided.

We have been corrected these dates and the period of COVID-19 cases occurred have been indicated, January-June 2020 (lines 167,175, and 180).

2.3% of participants were between the age range 0-4, and 14.2% between 5 and 14 (under legal minimum age in Spain). Could these participants adequately fill out the questionnaire survey? What procedure was followed?

When a child was chosen in the sampling, their parents were asked if they allowed their child to participate and answer the questionnaire with the help of the parents, considering the all clarifications before to perform the survey questionnaire. This is indicated in the text (lines 180-186). 

 How certain sociodemographic variables (e.g., social class) were calculated on the basis of the abovementioned fact?

The occupations of parents were asked in the questionnaire and the social class was estimated in two groups: Group I and II; professional, managerial and technical occupations; Group III-VI: skilled, non-manual or manual; partly-skilled; unskilled occupations. Children had the same social class of their parents (lines186-188). 

These aspects must be addressed as there is a considerable number of participants under 5 and 14 years old, thereby reflecting the need to provide more detailed description on procedure and methodological aspects.

We explain this aspect in the manuscript (lines 258-269).

Did the authors collect participants’ educational level? It should have been included as a control variable in the light of its connections with COVID-19 social behavior. 

We did not collect education level, we ask about occupation. In addition, the outbreak took place before the COVID-19 transmission was well known. 

The inclusion of each potential risk factor (confounding factors) should be previously explained/justified.

We use directed acyclic graphs to obtain a picture of the relationship between an exposure (mass gathering events) and an outcome (COVID-19 disease) and the factors, which have a role of confounders. An adjusted analysis of these factors was implemented. The factors are age, sex, social class, chronic illness, family COVID-19 case, and falla (social group); all these factors could modify the association between exposure (MGEs assistance) and outcome (COVID-19 disease) (lines 243-251).

Lines 419-421.

Fig. 3. Directed acyclic graphs (DAG) of mass gathering events (exposure) effect on COVID-19 disease (outcome). Ancestors of exposure and outcome (in red). Based on DAGitty version 3.0.

The section Analysis of COVID-19 outbreak and MGEs should be divided in to “smaller” sections so as to facilitate a clear picture of the current results.

We divide the section into smaller sections following the suggestion of the reviewer (lines 401,444,476, and 522).

The discussion section should be addressed considering prior comments.

We add some important issues, including two tables of infection rate and the basic reproductive number (Ro), more detail of the outbreak, and an epilog of this study (lines 564-570, and 591-647).

Table 9. Estimation of the basic reproductive number (Ro) from the MGE COVID-19 outbreak between 6 and 10 March 2020 plus 14 days

 Table 10. Attendance at pa-i-porta and Queen’s gala dinner MGEs and number of contacts (k) to obtain expected cases of COVID-19 compared with observed cases from the formula of Tupper and co-authors (63).

We have included new references of MEGs, how the MEGs are taking place now, and an epilog of our study: a cohort study of COVID-19 patients (lines 737-740). 

Certain grammatical and reading errors are still observed. This version must be proofread and edited by a native English-speaking expert.

The manuscript has been reviewed by a native English-speaking professor. 

 Castelló de la Plana, May 8, 2021

Trusting in your decision, best wishes.

 Dr. Alberto Arnedo-Pena, 

On behalf of the authors of the manuscript.

---

## [Decision Letter · Decision Letter 1]

16 Aug 2021

“MASS GATHERING EVENTS AND COVID-19 TRANSMISSION IN BORRIANA (SPAIN): A RETROSPECTIVE COHORT STUDY.”

PONE-D-21-04796R1

Dear Dr. Arnedo-Pena,

We’re pleased to inform you that your manuscript has been judged scientifically suitable for publication and will be formally accepted for publication once it meets all outstanding technical requirements.

Kind regards,

Simone Lolli

Academic Editor

PLOS ONE

Additional Editor Comments (optional):

Reviewers' comments:

Reviewer's Responses to Questions

**Comments to the Author**

1. If the authors have adequately addressed your comments raised in a previous round of review and you feel that this manuscript is now acceptable for publication, you may indicate that here to bypass the “Comments to the Author” section, enter your conflict of interest statement in the “Confidential to Editor” section, and submit your "Accept" recommendation.

Reviewer #1: All comments have been addressed

2. Is the manuscript technically sound, and do the data support the conclusions?

Reviewer #1: Yes

3. Has the statistical analysis been performed appropriately and rigorously? 

Reviewer #1: Yes

4. Have the authors made all data underlying the findings in their manuscript fully available?

Reviewer #1: Yes

5. Is the manuscript presented in an intelligible fashion and written in standard English?

Reviewer #1: Yes

6. Review Comments to the Author

Reviewer #1: (No Response)

7. PLOS authors have the option to publish the peer review history of their article (what does this mean?). If published, this will include your full peer review and any attached files.

Reviewer #1: **Yes: **Anna Wlodarczyk

---

## [Editor Report · Acceptance letter]

19 Aug 2021

PONE-D-21-04796R1 

“MASS GATHERING EVENTS AND COVID-19 TRANSMISSION IN BORRIANA (SPAIN): A RETROSPECTIVE COHORT STUDY.” 

Dear Dr. Arnedo-Pena:

I'm pleased to inform you that your manuscript has been deemed suitable for publication in PLOS ONE. Congratulations! Your manuscript is now with our production department. 

Kind regards, 

on behalf of

Dr. Simone Lolli 

Academic Editor

PLOS ONE